# Modulating p38 MAPK signaling by proteostasis mechanisms supports tissue integrity during growth and aging

Wang Yuan [1], Yi M. Weaver[1], Svetlana Earnest[1], Clinton A. Taylor IV[1], Melanie H. Cobb [1] & Benjamin P. Weaver [1] ✉

The conserved p38 MAPK family is activated by phosphorylation during stress responses and inactivated by phosphatases. *C. elegans* PMK-1 p38 MAPK initiates innate immune responses and blocks development when hyper-activated. Here we show that PMK-1 signaling is enhanced during early aging by modulating the stoichiometry of non-phospho-PMK-1 to promote tissue integrity and longevity. Loss of *pmk-1* function accelerates progressive declines in neuronal integrity and lysosome function compromising longevity which has both cell autonomous and cell non-autonomous contributions. CED-3 caspase cleavage limits phosphorylated PMK-1. Enhancing p38 signaling with caspase cleavage-resistant PMK-1 protects lysosomal and neuronal integrity extending a youthful phase. PMK-1 works through a complex transcriptional program to regulate lysosome formation. During early aging, the absolute phospho-p38 amount is maintained but the reservoir of non-phospho-p38 diminishes to enhance signaling without hyperactivation. Our findings show that modulating the stoichiometry of non-phospho-p38 dynamically supports tissue-homeostasis during aging without hyper-activation of stress response.

The p38 family consists of stress-sensing MAPKs that respond to diverse stimuli including chemical and physical stressors, microbial infections, as well as cytokine cues[1]. Licensing of p38 MAPK signaling pathways following physiological stressors relies on dual phosphorylation of the highly conserved TGY motif in the activation loop by upstream activating kinases. Recently, it was shown that both pathogenic and non-pathogenic stresses initiate multimerization of the membranous Toll/interleukin-1 receptor domain protein (TIR-1) receptor ultimately leading to activation of the PMK-1 immunity program in intestinal cells[2]. Molecular docking simulations indicated the mammalian TIR, TIRAP, binding of p38 is dependent on its own phosphorylation states[3]. These studies indicate the importance of upstream inflammatory responses enhancing p38 activation. Balancing this activation, phosphatases including dual specificity phosphatases (DUSPs) dephosphorylate the TGY motif, making for cycles of activation and deactivation[4]. In the case of *C. elegans* PMK-1 p38 MAPK, VHP-1 is the specific DUSP that limits PMK-1 phosphorylation[5,6].

Stress response pathways have complex effects on aging. Longevity requires the integration of many processes including resolving protein aggregates, nutrient sensing, and genomic integrity among other factors[7]. PMK-1 was previously shown to activate the majority of pathogen response genes via the transcription factor ATF-7[8] and the PMK-1-ATF-7 circuit is a key determinant for longevity in response to nutrient availability or infection[9]. Activation of SKN-1 (NRF) by PMK-1 is required to ensure longevity by maintaining proper redox balance[10,11]. Loss of the PMK-1-dependent pathogen-response program results in immunosenescence with aging, thereby shortening life span when animals are exposed to pathogenic bacteria[12,13]. Additionally, PMK-1 was recently shown to activate ubiquitination, SUMOylation and neddylation factors in response to bacterial infection[14].

Caspases have been identified as key regulators of non-apoptotic cell fate during metazoan development. Muscle cell differentiation in mice requires chromatin reorganization by elimination of PAX7 and SATB2 by caspases-3/7[15,16]. Pluripotency potential is limited in an

[1]Department of Pharmacology, UT Southwestern Medical Center, Dallas, TX, USA. ✉e-mail: Benjamin.Weaver@UTSouthwestern.edu

epidermal stem cell lineage in *C. elegans* by CED-3 caspase and UBR-1 E3 ligase-mediated elimination of LIN-28[17,18]. Asymmetric cellular fates for neuroblasts in *C. elegans* is determined by CED-1 (MEGF10 receptor) and CED-3 caspase antagonizing PIG-1 (MELK)-dependent mitotic potential[19]. Intestinal progenitor cell quiescence in *Drosophila* is determined by Dronc caspase regulation of Notch signaling[20]. These findings underscore the contribution of caspases working outside of cell death to ensure cell fate decisions. In contrast to caspase-mediated functions in development, much less is known about possible roles of caspase-mediated regulation of signaling in aging.

Proteostasis involves a complex network of factors ensuring native protein functions while also mitigating aberrant protein functions and this network is progressively challenged during aging[21-23]. The function of proteostasis in mitigating stress responses such as unfolded protein responses have clear consequences on longevity and degenerative diseases[24-27]. Long-lived mutants have enhanced proteostasis control including clearance of misfolded proteins[28].

In addition to chaperone-mediated protein folding, proteostasis clearance pathways including the ubiquitin-proteasome system as well as the autophagy-lysosome pathway are critical to support the proteostasis network. Disruption of these degradation systems compromises health and lifespan[25]. The proteostasis network is best understood for its role in mitigating stressful or pathological states such as the heat shock response which requires integration of signaling between cells to sense and respond accordingly[25]. However, the impact of proteostasis on cell signaling pathways and its effects on normal tissue aging remains a major area for further understanding.

We recently reported that CED-3 caspase antagonizes a PMK-1-dependent anti-microbial function to promote post-embryonic development in *C. elegans*[29]. PMK-1 hyperactivation is critical for survival in the presence of pathogens or abiotic stressors but can become detrimental leading to stalled development[5,29-31]. Altogether, these findings suggest that p38 MAPKs may have broader functions in integrating cross-talk between stress responses and homeostatic programs. In this study, we investigated the role of PMK-1 signaling in maintaining tissue integrity during aging in the absence of stress. We find that PMK-1 signaling is enhanced during early aging and this function is protective to pain sensory neurons and promotes longevity. Surprisingly, the enhancement of this PMK-1 signaling results from modulating the stoichiometry of non-phospho-PMK-1. We further show that the well-established stalling of development by hyperactivation of p38 MAPK can be ameliorated by expanding the non-phospho PMK-1 pool without impacting accumulation of phospho-PMK-1.

## Results

### PMK-1 modulated by CED-3 caspase cleavage regulates lifespan

We previously identified *C. elegans* PMK-1 as a proteolytic substrate of CED-3 caspase using in vitro cleavage assays[29]. The CED-3 cleavage site is found among multiple p38 members from diverse animal phyla and located just beyond the core kinase domain at the beginning of the C-terminus (CT), which is highly conserved throughout metazoans (Fig. 1a-b). Alphafold models of *C. elegans* PMK-1 and human p38γ show that the cleavage site is solvent accessible suggesting this conserved region is exposed for caspase access (Supplementary Fig. 1a). The C-terminus in MAPK has been shown to be important for promoting active MAPK conformations, suggesting that cleavage within this region will impair p38 function.

In previous studies, loss of *pmk-1* function has been shown to compromise longevity during exposure to pathogenic bacteria but no effect on aging was revealed on non-pathogenic food when germline proliferation was blocked by treatment with FUDR[9,12]. We tested the

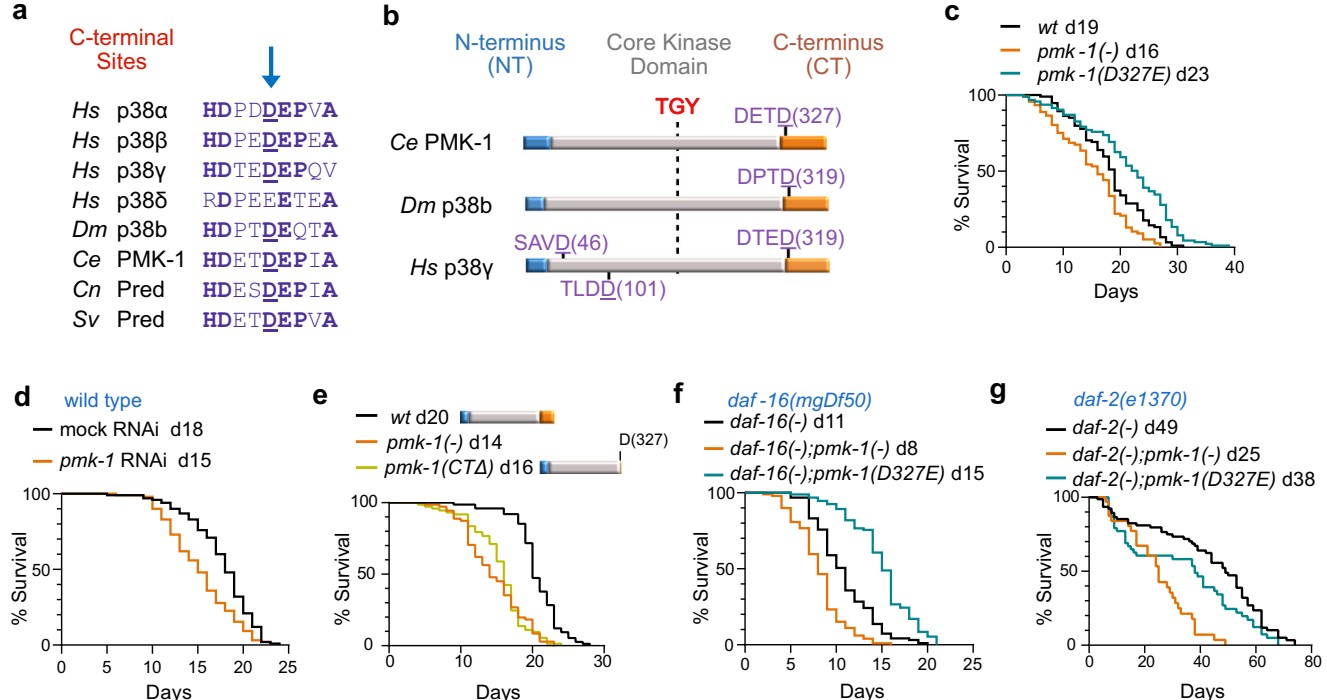

**Fig. 1 | Caspase cleavage of p38 MAPK PMK-1 controls longevity in *C. elegans*.** **a** Alignment of p38 homologs putatively conserved cleavage site (arrow) at the C-terminus. *Hs, H. sapiens; Dm, D. melanogaster; Ce, C. elegans; Cn, C. nigoni; Sv, S. vulgaris;* Pred, Predicted PMK-1 homologs. **b** A diagram showing p38 dual phospho-activation sequence (TGY) and predicted cleavage sites (DXXD). **c** Aging assay of wild type (wt), *pmk-1(-)* null and caspase cleavage resistant *pmk-1(D327E)* mutant animals. **d** Aging assay of wild type animals with *pmk-1* RNAi. **e** Aging assay of animals with PMK-1 C-terminal deletion *pmk-1(CTΔ)*. **f** Effects of *pmk-1(-)* null and caspase cleavage resistant *pmk-1(D327E)* mutations on *daf-16* (foxo) null mutant aging. **g** Effects of *pmk-1(-)* null and caspase cleavage resistant *pmk-1(D327E)* mutations on *daf-2* (insulin receptor) null mutant aging. **c–g** Median survival days indicated following letter d in the graph legend. Additional independent replicates of aging assays and statistics are shown in Supplementary Fig. 1 and Data 1. Source data are provided as a Source Data file.

effects of the *pmk-1* null mutant on aging when fed normal food (no FUDR) allowing for normal germline proliferation. We found that loss of *pmk-1* function had shortened adult lifespan under normal dietary conditions (Fig. 1c). Our results differ significantly from the previous findings when FUDR was used to stop germline proliferation. This difference is consistent with the tradeoff of fecundity versus proteostasis in determining lifespan[28,32].

To abolish caspase cleavage of PMK-1, we used a *pmk-1(D327E)* cleavage-resistant mutation in the endogenous locus using CRISPR mutagenesis and outcrossed this strain multiple times (Methods)[29]. Changing aspartate to glutamate in this position abolishes cleavage[29] but preserves charge. Moreover, glutamate is naturally occurring at this position in some p38 members and many MAPK homologs (Fig. 1a). In contrast to our findings of short-lived *pmk-1* null mutants (Fig. 1c), the caspase cleavage-resistant *pmk-1(D327E)* mutants had an extension of lifespan (Fig. 1c; Supplementary Fig. 1b; Supplementary Data 1). Consistent with the *pmk-1* null, wild-type animals fed *pmk-1* RNAi had shortened lifespan compared to mock RNAi treated animals (Fig. 1d, Supplementary Fig. 1c; Supplementary Data 1). Loss of the PMK-1 C-terminus (CTΔ) causes shortened lifespan similar to the *pmk-1(-)* null mutation suggesting that the cleaved protein is not functional (Fig. 1e and Supplementary Data 1). In addition to acting on PMK-1, CED-3 caspase has many other critical functions including germline apoptosis during adulthood. Thus, it is not surprising that we observed *ced-3* null mutants had diminished life span (Supplementary Fig. 1b and Supplementary Data 1).

Insulin signaling is essential for metabolic adaptation and basal stress resistance, functioning as a major determinant of longevity in nematodes[33,34], flies[35,36] and mice[37,38]. To determine the relationship between PMK-1 signaling and insulin-dependent longevity, we tested how *pmk-1* null and *pmk-1(D327E)* mutations impact the lifespans of *daf-16* (FOXO transcription factor) and *daf-2* (insulin receptor) mutants. We found that the cleavage-resistant *pmk-1(D327E)* mutation extended the lifespan of *daf-16*-deficient mutants, whereas *pmk-1* null mutants further reduced lifespan in animals with compromised *daf-16* function (Fig. 1f; Supplementary Fig. 1d and Supplementary Data 1). Extending lifespan of *daf-16* mutants by *pmk-1(D327E)* mutation suggests that a key sub-set of *pmk-1*-dependent genes act through additional transcriptional pathways supporting longevity. The *pmk-1(D327E)* mutation did not restore *daf-16* mutants to normal lifespan suggesting that PMK-1 works partially through DAF-16 (Supplementary Fig. 1e and Supplementary Data 1). Interestingly, both *pmk-1* null mutation and the cleavage-resistant mutation reduced longevity of *daf-2* deficient mutants (Fig. 1g and Supplementary Data 1). The reduction of *daf-2* longevity by *pmk-1* null mutation is consistent with previous work demonstrating that *daf-2*-enhanced pathogen resistance required intact *pmk-1*[12]. However, reduction of *daf-2* lifespan by *pmk-1(D327E)* suggests the possibility that heightened PMK-1 function can partially restore *daf-2* signaling.

## PMK-1 gene expression programs support tissue homeostasis

To understand how *pmk-1* null mutation and the cleavage-resistant *pmk-1(D327E)* mutation inversely impact longevity, we examined their effects on global gene expression programs using mRNA-seq (Fig. 2a and Supplementary Data 2). We observed that 76% of genes altered by the *pmk-1* null mutation were inversely regulated by the cleavage-resistant *pmk-1(D237E)* mutation (Fig. 2a). We then performed enrichment analysis for genes altered by loss of *pmk-1* function. Consistent with previous findings[12,13], loss of *pmk-1* resulted in a significant decline in innate immunity genes (Fig. 2b). Intriguingly, we also found altered expression of lytic vacuole genes, extracellular components, such as collagens and proteases, posterior ventral process D (PVD) sensory neuron genes, as well as intestine genes (Fig. 2b).

We examined the PVD sensory neuron and lysosomal gene sets further for their effects on expression with and without functional

*daf-16*(FOXO). Most genes altered by loss of *pmk-1* are also affected by loss of *daf-16*, suggesting PMK-1 signaling and DAF-16 co-regulate these genes. However, the inverse effects of *pmk-1(-)* null versus *pmk-1(D327E)* for both PVD (Fig. 2c) and lysosomal genes (Fig. 2d) were not grossly altered by *daf-16* status, indicating there are other transcription factors in this regulatory network. Another *daf-16* mutation had comparable findings (Supplementary Fig. 2a, b). Importantly, loss of *pmk-1* resulted in dysregulation for both neuronal and lysosomal genes including some genes upregulated while others were down-regulated (Fig. 2c, d). We were particularly interested in these results given the importance that maintaining proper stoichiometry of organelle components has on maintaining proteostasis.

Based on our findings that *daf-16* was not strictly required for PMK-1 regulation of lysosomal and neuronal genes, we examined the proximal promoters of the inversely regulated genes to identify potential other transcription factor binding site enrichments (Fig. 2e and Supplementary Data 3). These enrichments were then compared to PMK-1-dependent PVD neuronal, innate immunity, and extracellular matrix genes (Fig. 2e). DAF-16 was among the top enriched TFs, consistent with the mRNA-seq results that many lysosomal genes that altered by loss of *pmk-1* are also affected by loss of *daf-16*. In addition to DAF-16, we also observed a dozen more TFs that have binding sites for more than 30% of lysosomal genes that were altered by loss of *pmk-1*. Altogether, our findings predict a model whereby PMK-1 p38 MAPK signaling works with a network of transcription factors to integrate inputs from other pathways such as insulin signaling to regulate gene expression programs.

In addition to gene expression, we investigated how the *pmk-1* null and *pmk-1(D327E)* cleavage-resistant mutation impact protein clearance by the proteasome degradation pathway. We enriched for poly-ubiquitin conjugated proteins following a brief treatment with Bortezomib to accumulate proteasome substrates. We found that total K48-linked protein accumulation was comparable amongst all strains (Supplementary Fig. 2c–e). We then identified poly-ubiquitin conjugated proteins by mass-spectrometry and analyzed their alterations in all *pmk-1* mutants (Fig. 2f and Supplementary Data 4). We found that more than 80% poly-ubiquitin conjugated proteins altered in *pmk-1* null were also similarly altered in *pmk-1(CTΔ)* mutants (Fig. 2f). This finding paralleled their similar phenotypes in reduced longevity.

The poly-ubiquitin conjugated proteins were then analyzed for GO term enrichments using the Wormbase enrichment toolkit. We found several cellular processes inversely targeted for degradation in the *pmk-1(-)* null versus the cleavage-resistant *pmk-1(D327E)* mutants including extracellular matrix structural constituent, unfolded protein binding, peptide synthesis and ATP hydrolysis (Fig. 2g, h, Supplementary Fig. 2f). Altogether, our findings suggest that the cleavage-resistant *pmk-1(D327E)* mutant behaves oppositely to the *pmk-1* null mutant and has heightened p38 signaling in transcriptional programs. In addition to innate immunity, we find p38 signaling regulates both gene expression and degradation programs underlying tissue homeostasis.

## PMK-1 regulates lysosome formation and activity

The lysosomal branch of protein clearance is key to support proteostasis and tissue dynamics[39–42]. Long-lived mutants have recently been shown to retain young adult lysosome morphology as they progress into aging[43]. Because we observed the altered lysosomal gene expression by loss of *pmk-1*, we further investigated if altered lysosomal function contributes to shortened lifespan of *pmk-1(-)* mutants. Lysosomes function in complex organelle trafficking as they fuse with a variety of membrane-bound cargo. After lysosome fusion with the autophagosome, the autolysosome resolves cargo and subsequent tubulation regenerates lysosomes upon scission in a multi-step process[44,45]. Tubules are thought to represent autolysosome extensions and can indicate a compromised ability to reform lysosomes as

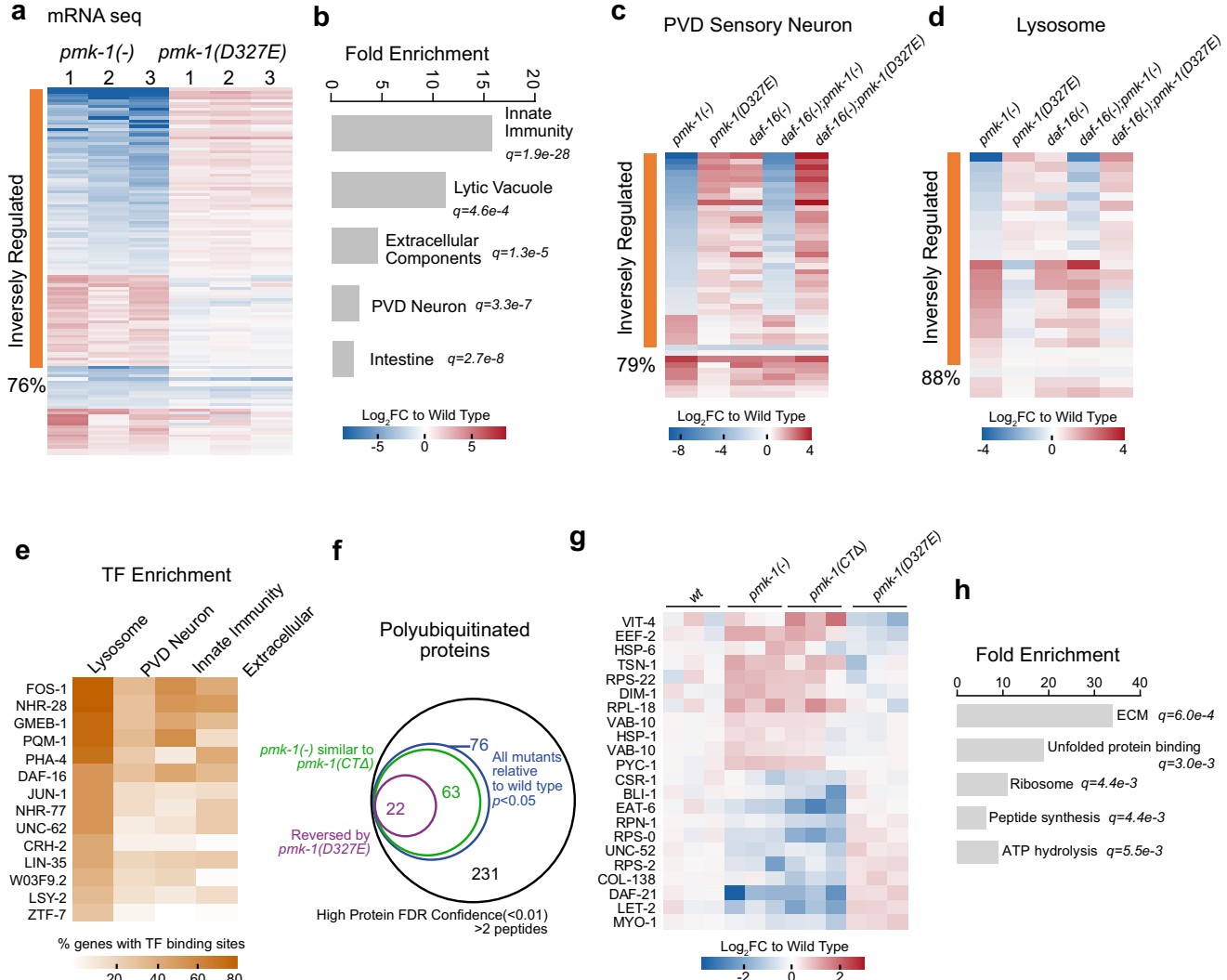

**Fig. 2 | PMK-1 signaling is regulated by caspase cleavage supporting tissue homeostasis. a** mRNA-Seq analysis to identify *pmk-1*-dependent genes regulated by caspase cleavage in day 1 young adults. *n* = 3 biological replicates for each genotype. FDR < 0.1 for *pmk-1(-)*. Log2-fold change of each replicate for *pmk-1(-)* and *pmk-1(D327E)* compared to wt was shown as heat map. **b** Pathway enrichment of PMK-1 regulated genes. **c-d** *pmk-1*-dependent sensory neuron and lysosome genes for *daf-16* dependence. PMK-1-dependent genes clustered by inverse regulation with PMK-1(D327E) and plotted with Log2-fold change compared to wild type. n = 2 biological replicates. **e** Frequency of transcription factor binding sites in

proximal promoters of *pmk-1* dependent genes (Datasource: Wormbase). **f** Diagram depicting overlapping of change in polyubiquitinated protein shared between *pmk-1(-)* and *pmk-1(CTΔ)* compared to wild type. *n* = 3 biological replicates. **g** Heatmap of Log₂ Fold change of polyubiquitinated proteins that are inversely regulated by *pmk-1(-)* and *pmk-1(D327E)* compared to wild type. *n* = 3 biological replicates. **h** Pathway enrichment of polyubiquinated proteins that are inversely regulated by *pmk-1(-)* and *pmk-1(D327E)*. **b**, **h** Wormbase GeneSet Enrichment tool. Source data are provided as a Source Data file.

they progressively elongate[45], consistent with failure to mitigate proteostasis challenges with aging.

Using the $P_{CED-1}$::*nuc-1*::mCherry marker which localizes the NUC-1::mCherry fusion protein inside epidermal lysosomes[43], we monitored lysosome morphology during aging. For comparable quantitation, we imaged the same anatomical location across all animals and quantified all puncta and tubules in the views. At the onset of adulthood (day 1), lysosomes in wild-type animals were mostly particulate, with very few visible tubules (Fig. 3a). Following egg-laying (day 5) during early aging, wild-type animals displayed increased tubular lysosome structures (Fig. 3a, b). We observed that *pmk-1* null and *pmk-1(CTΔ)* mutants had enlarged lysosome structures at the onset of adulthood (day 1) and continued with aging (day 5) (Fig. 3a, b and Supplementary Fig. 3a, b). In contrast, cleavage-resistant *pmk-1(D327E)* mutants maintained youthful lysosomes with smaller particle size and much reduced tubular formation on day 5 of adulthood

compared to wild-type (Fig. 3a, b). Elimination of PMK-1(D327E) protein using an auxin-induced degron (AID) tag resembled both the *pmk-1(-)* null and *pmk-1(CTΔ)* mutants with enlarged lysosome structures (Supplementary Fig. 3c), further confirming that the opposite lysosomal phenotypes are *pmk-1*-dependent. It has been shown that long lived mutants such as *daf-2* null animals maintain very uniform particle size with few tubular structures[43] similar to our observation for *pmk-1(D327E)* mutants. Intriguingly, wild-type and *ced-3* null mutants displayed comparable average particle size and tubular lysosome structures on day 1 and day 5 of adulthood (Fig. 3b). However, *ced-3* null mutants displayed increased variability in both particle and tubular structures. These results suggest that in addition to PMK-1, the caspase may act on additional factors that affect lysosome function.

The lysosome enlargement for *pmk-1* null animals is progressive from the first (L1) to the final (L4) larval stages (Fig. 3c). To examine

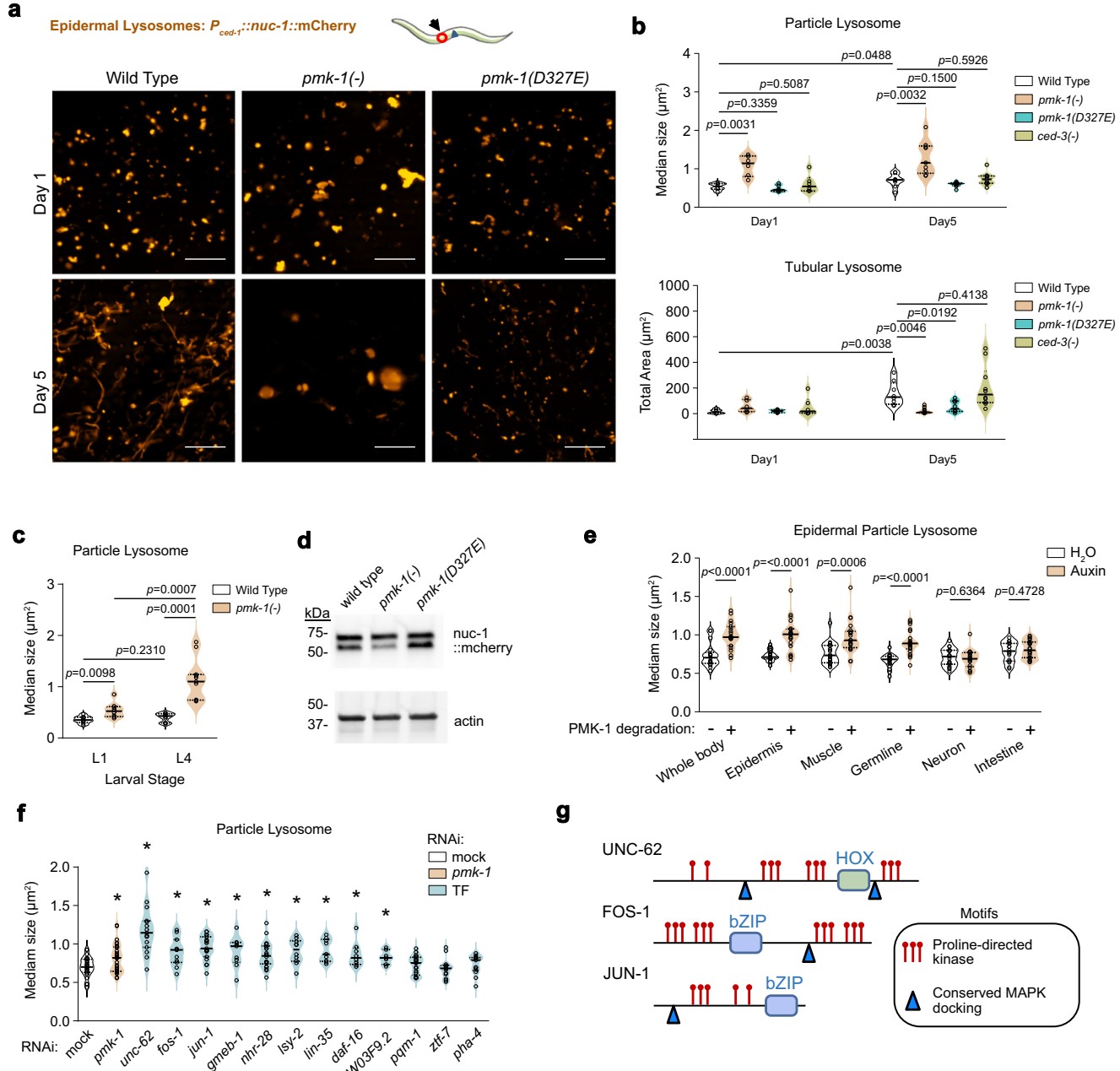

**Fig. 3 | Cell non-autonomous PMK-1 signaling is required for lysosome formation and activity. a, b** Pseudo-colored epidermal lysosome marker ($P_{CED-1}$::NUC-1::mCherry) and quantification of particle lysosome size and total tubular lysosome area on Day 1 and Day 5 adulthood. Scale bar, 10 μm. Arrow and red circle indicate anatomical position of imaging. All particles and tubules in view of all images were plotted as median area (particles) or total area (tubules). Each open circle dot within the violin plot represents median value of all particles imaged in 1 animal or total area of tubules in 1 animal. Day1, all genotypes $n = 6$ animals. Day5, wt n = 8, *pmk-1(-)* $n = 9$, *pmk-1(D327E)* $n = 8$, *ced-3(-)* $n = 10$ animals. **c** First (L1) and final (L4) larval stage epidermal lysosome particle size quantification. WT L1 $n = 9$, wt L4 $n = 10$, *pmk-1(-)* L1 $n = 8$, *pmk-1(-)* L4 $n = 11$ animals. **d** Western blot of NUC-1::mCherry protein showing *pmk-1*-dependent processing. Data from one experiment with hundreds of worms per sample. **e** Lysosome particle sizes of day 1 adult with tissue-specific elimination of PMK-1 protein. Data pooled from two

independent experiments. Whole body±degradation $n = 17/19$; Epidermis±degradation $n = 20/20$; Muscle±degradation $n = 22/21$; Germline±degradation $n = 18/22$; Neuron±degradation $n = 19/17$; Intestine±degradation $n = 16/20$. **f** RNAi test of transcription factors identified by promoter analysis (Fig. 2e) on epidermal lysosome particle size. Data pooled from three independent experiments. Each experiment includes a subset of transcription factors. Mock and *pmk-1* RNAi performed each time as negative and positive controls. Mock: $n = 27$, *pmk-1*: $n = 33$, *unc-62*: $n = 15$, *fos-1*: $n = 11$, *jun-1*: $n = 19$, *gmeb-1*: $n = 9$, *nhr-28*: $n = 22$, *lsy-2*: $n = 10$, *lin-35*: $n = 9$, *daf-16*: $n = 8$, *W03F9.2*: $n = 10$, *pqm-1*: $n = 21$, *ztf-7*: $n = 14$, *pha-4*: $n = 13$ animals. **g** Motif analysis for Proline-directed phosphorylation sites and conserved MAPK docking of TFs. **b,c,e,f** $p$ value from unpaired, two-tailed Welch's t-test. **f** asterisk, $p < 0.05$. Exact $p$ values for **f** provided in Source data. Source data are provided as a Source Data file.

functional effects on lysosome proteolytic processing, we monitored NUC-1::mCherry processing for Day 1 adults and found that loss of *pmk-1* function had reduced processing (Fig. 3d). The findings of lysosomal morphology and function parallel our observations that

*pmk-1* null mutants have shortened life spans whereas *pmk-1(D327E)* mutants display prolonged life spans.

We examined whether the defective enlargement of lysosomes was cell autonomous for PMK-1 signaling using a series of

tissue-specific TIR1 E3 ligase mutants that target the auxin-inducible degron on the N-terminus of PMK-1 (Supplementary Fig. 4). Tissue-specific expression of TIR1 E3 ligase was achieved using promoters with defined expression patterns (Supplementary Fig. 4b)[46,47]. Whole body elimination of PMK-1 function showed enlarged lysosomes at day 1 adulthood (Fig. 3e) similar to our findings for the *pmk-1* null mutant. Loss of PMK-1 specifically in epidermis also showed enlarged epidermal lysosomes (Fig. 3e) suggesting that lysosome trafficking has a *pmk-1*-dependent cell-autonomous component.

We further tested other tissues and revealed cell non-autonomous p38 signaling for lysosome formation. Consistent with our findings that PMK-1 is expressed in multiple tissues (Supplementary Fig. 4), we found that PMK-1 functions in muscle and germline are required for epidermal lysosome formation. Because lysosomes are a major clearance pathway for proteostasis, our results indicate that epidermal proteostasis is supported directly within the epidermis as well as by distal tissues including the germline and muscle. Our findings that germline PMK-1 supports organismal proteostasis provided a potential mechanism for the observation that loss of *pmk-1* results in shortened lifespan with germline proliferation.

Based on our identification of transcription factors (TFs) enriched for binding sites in lysosomal genes (Fig. 2e), we tested the top 12 TFs for defects in lysosomal morphology. We found that loss of the Meis homeobox ortholog *unc-62* by RNAi had the most significant increase in epidermal lysosome particle size (Fig. 3f). Loss of seven other transcription factors by RNAi including *fos-1* (FOS), *jun-1* (JUN), *gmeb-1* (GMEB), *nhr-28* (HNF4), *lsy-2* (ZNF18), *lin-35* (RBL) and *daf-16* (FOXO) also had significantly increased lysosome particle size (Fig. 3f), suggesting that PMK-1 signals a complex transcriptional network to regulate lysosomal genes.

The p38 MAPK family mediates signaling by phosphorylation of serine or threonine residues on target proteins with a canonical proline-directed motif (S/TP) and substrates typically contain a MAPK-docking motif. Based on consensus sequences, UNC-62, FOS-1, and JUN-1 are enriched for proline-directed motifs and each contain at least 1 docking motif (Fig. 3g and Supplementary Fig. 5a). We analyzed strains bearing GFP fusions of these transcription factors[48,49] (Supplementary Data 5) for alterations in total protein or extent of phosphorylation (Methods). When treated with *pmk-1* RNAi, we observed alterations of phospho-forms or total protein accumulation for all three transcription factors (Supplementary Fig. 5b). Based on the structural motifs along with alterations in phospho-status or protein stability, it is conceivable that these transcription factors are directly regulated by a MAPK signaling cascade.

## PMK-1 signaling supports neuronal integrity during aging

Lysosomes are essential for organismal proteostasis. Neurons are particularly sensitive to declines in organismal proteostasis and succumb to deficits in cellular clearance pathways. Loss of neuronal integrity is a significant feature of aging throughout animal phyla and aging modulates the trajectory of neurodegenerative processes[50–53]. Because our gene expression data also revealed significant enrichments for neuronal genes regulated by PMK-1, we evaluated aging phenotypes associated with neurons in *C. elegans*. Using a pan-neuronal marker, $P_{RGEF-1}$::DsRed[54], we found that *pmk-1* null, *pmk-1(D327E)*, and *ced-3* null mutants had normal neuronal morphology at young adulthood (day 1) (Supplementary Fig. 6a, b). By day 8 of adulthood, about half of wild-type animals had disrupted morphology in the large lateral mid-body sensory neurons with obvious puncta demonstrating a progressive decline in neuronal integrity with aging (Fig. 4a, b). Strikingly, this decline in neuronal integrity was accelerated in the majority of *pmk-1* null and *pmk-1(CTΔ)* mutants as well as *pmk-1* RNAi of wild-type animals by day 8 of adulthood (Fig. 4a, b and Supplementary Fig. 6c–e). In contrast, we saw that both the cleavage-resistant *pmk-1(D327E)* and *ced-3*

mutants were protective for neuronal morphology with aging (Fig. 4a, b).

Using a reporter specific for pain sensory neurons (posterior ventral process D, PVD), we confirmed that loss of *pmk-1* function led to an accelerated decline in integrity of the PVD sensory neurons with obvious puncta whereas the cleavage-resistant *pmk-1(D327E)* mutation showed a strong protective affect for neuronal integrity with few puncta, long continuous tracts and branching (Fig. 4c, d). These findings suggest that although p38 MAPK signaling is not required for development of these neurons it is a key regulator that can be modulated to protect neuronal integrity with aging.

We examined cell autonomy of PMK-1 in supporting PVD neuronal integrity. Using the series of tissue-specific TIR1 E3 ligase-expressing strains to target the auxin-inducible degron on the N-terminus of PMK-1, we found that loss of PMK-1 function in the whole body, the epidermis and the germline had significant declines in PVD neuronal integrity (Fig. 4e). Revealing key functions for epidermal and germline PMK-1 signaling in supporting PVD neuronal integrity was similar to our findings for lysosome function. It is further intriguing that the PVD sensory neurons are anatomically connected with the epidermis. Previous work in *C. elegans* established the importance of extruding toxic protein aggregates and damaged mitochondria as exophers to support touch neuron proteostasis during neurotoxic stress and that the hypodermis (epidermis) is important for recognition and degradation of the extruded exophers[55]. In our case, disruption of p38 signaling in the epidermis compromising the epidermal lysosomes may contribute to compromising the anatomically-associated sensory neurons in a similar fashion.

To examine the functional impact of PMK-1 signaling on PVD neurons during aging, we utilized a harsh-touch PVD-neuron dependent behavioral assay. At day 1 of adulthood, all animals correctly reversed their course in response to harsh-touch (Fig. 4f). This is consistent with our findings that PVD neurons form normally in *pmk-1* null mutants (Supplementary Fig. 6a, b). By day 8 of adulthood, we found that wild-type animals had a strong reduction in harsh-touch reversal (Fig. 4f). This trend was accelerated in *pmk-1* null mutants. In contrast, *pmk-1(D327E)* and *ced-3(-)* null mutants demonstrated an extended phase of youthful responsiveness (Fig. 4f).

## Phospho-p38 maintained but non-phospho-p38 dynamic in aging

Because we observed that *pmk-1(D327E)* mutant had heightened PMK-1 signaling in gene expression supporting longevity, we asked how CED-3 cleavage of PMK-1 enhances PMK-1 signaling. We first tested whether *pmk-1(D327E)* mutants have increased PMK-1 phosphorylation (pPMK-1). To examine protein expression during development and aging, we used CRISPR mutagenesis to add an HA tag at the N-terminus of both endogenous PMK-1 and cleavage-resistant PMK-1(D327E). Adding tags at the N-terminus of PMK-1 does not affect function as determined by induction of a *pmk-1*-dependent reporter and comparable expression profiles (Supplementary Fig. 7a).

Surprisingly, the amount of pPMK-1(D327E) was similar to wild-type pPMK-1 during both larval development and aging (Fig. 5a–d and Supplementary Fig. 7b, c). However, total PMK-1(D327E) protein was reduced by ~50% at each stage of larval development and throughout aging compared to wild-type (Fig. 5a–d and Supplementary Fig. 7b, c). Steady state *pmk-1* mRNA was comparable between wild-type and the cleavage-resistant mutant (Supplementary Fig. 7d).

The observation that pPMK-1 was kept relatively constant between the wild-type and cleavage resistant mutants suggest that the absolute amount of pPMK-1 in an animal may be strictly regulated in the absence of stress. However, the reduction of total PMK-1(D327E) protein accompanied with enhanced PMK-1 signaling suggested the possibility that non-phosphorylated PMK-1 could impact net signaling. The strict regulation of pPMK-1 was also observed in wild type animals

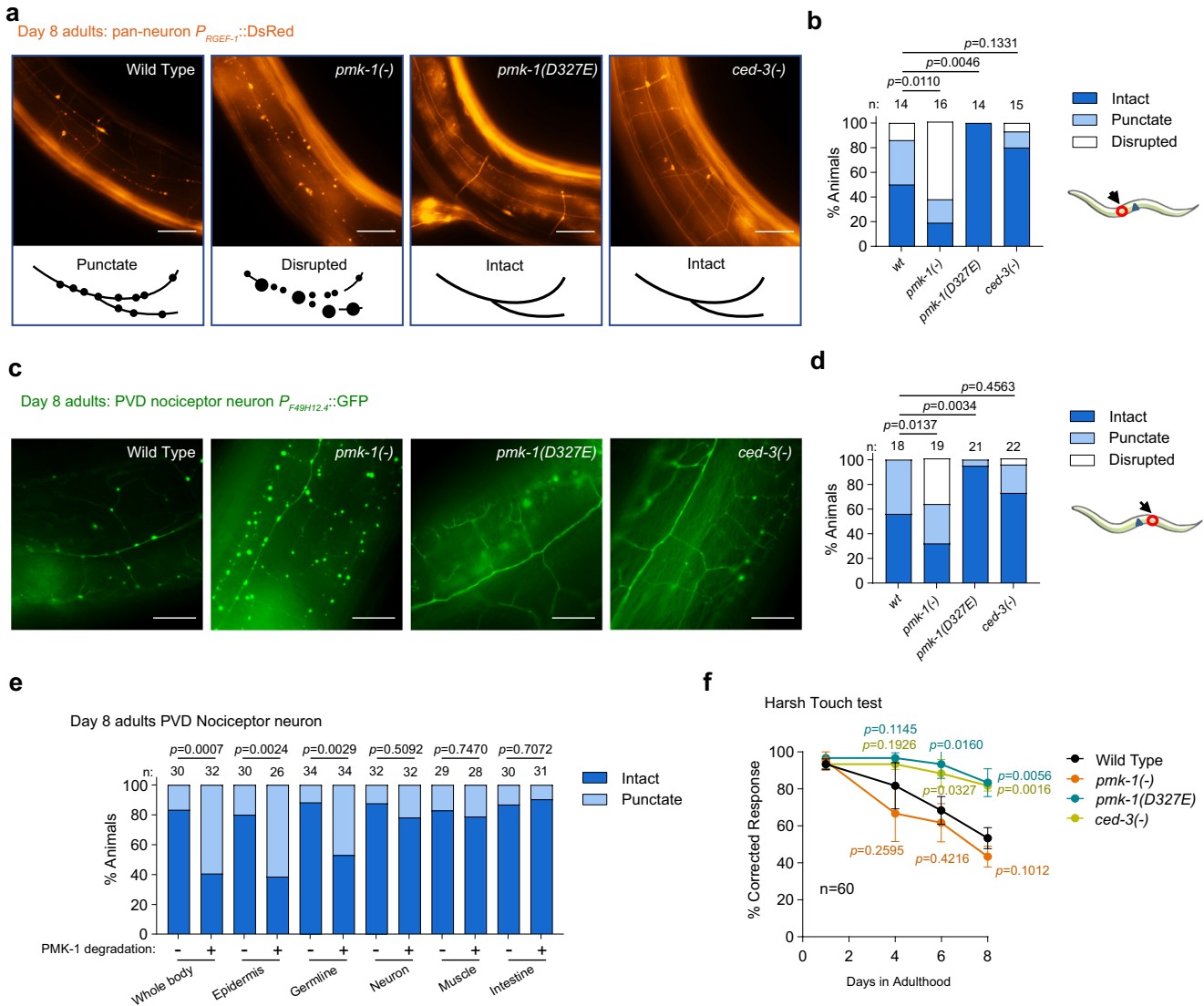

**Fig. 4 | Caspase cleavage of PMK-1 regulates integrity of sensory neurons during aging. a, b** Representative images and quantification of large lateral mid-body sensory neurons during aging depending on PMK-1 signaling regulated by caspase cleavage. Neurons were visualized using a pan-neuronal marker ($P_{RGEF-1}$::DsRed, pseudo color). Scale bar, 50 μm. Continuous tracts (intact), nerves with punctate foci (punctate), and disrupted nerves with large foci (disrupted). Arrow and red circle indicate anatomical position of imaging. n, number of animals. Neuron integrity *p* value from Chi-square test for trend. **c, d** Representative images and quantification of PVD sensory neurons visualized using a PVD/AQR-specific marker (*F49H12.4::GFP*, pseudo color). Scale bar, 20μm. **e** Quantification of PVD neuronal integrity with tissue-specific elimination of PMK-1. *n* number of animals pooled from two independent experiments. *p* value from Fisher's exact test. **f** Harsh-touch behavioral assay. Animals scored for course reversal following harsh-touch stimulus. dot, mean value of 3 independent experiments with *n* = 20 animals per experiment. Data are presented as mean values ± SD. *p* value from two-tailed, unpaired t-test. Source data are provided as a Source Data file.

during aging. We found that the amounts of pPMK-1 were relatively constant throughout adulthood (pPMK-1, Fig. 5d) whereas total PMK-1 markedly diminished with aging (Total PMK-1, Fig. 5d).

Using mass-spectrometry, we find that the phosphorylated TGY-containing peptide of the activation site (Fig. 5e) for PMK-1 (QTDSEMTGYVATR) is limited to nearly 1 out of 20 peptides for that sequence indicating that pPMK-1 was a small portion of total PMK-1 pool at Day 1 adulthood in the absence of stress. As animals age, the total pool of PMK-1 is diminishing rapidly whereas the small amount of pPMK-1 was maintained.

**Phospho-p38 pool limited by caspase degraded by proteasome**
Finding that animals tightly regulate pPMK-1 levels throughout adulthood suggests robust regulatory mechanisms of pPMK-1 in vivo. We then considered the possibility that the down regulation of PMK-1(D327E) total protein (Fig. 5b, d) was a compensatory mechanism to

maintain a constant amount of pPMK-1 in vivo. To overcome adaptation, we first generated an overexpression system controlled by auxin-induced degron (AID) for transient overexpression. We used CRISPR mutagenesis to add a second copy of *pmk-1* in a MosI site (Methods). The extra *pmk-1* copy was tagged with HA-AID and driven by a constitutive promoter (*eft-3*). For wild-type overexpression (WT), we used full length *pmk-1* open reading frame with intact exons and introns. For cleavage-resistant PMK-1 (DE), we made a D327E mutation in the same ORF before MosI insertion. To overcome adaptive compensation, we maintained these strains on auxin plates to prevent overexpression.

Placing synchronous L1 animals on normal food without auxin to allow overexpression, we found that the cleavage-resistant mutation was able to increase the amount of phosphorylated PMK-1 about 10-fold compared to the wild-type overexpression (Fig. 5f, g and Supplementary Fig. 8a). We then tested if this compensation happens at the endogenous locus. Similar results were

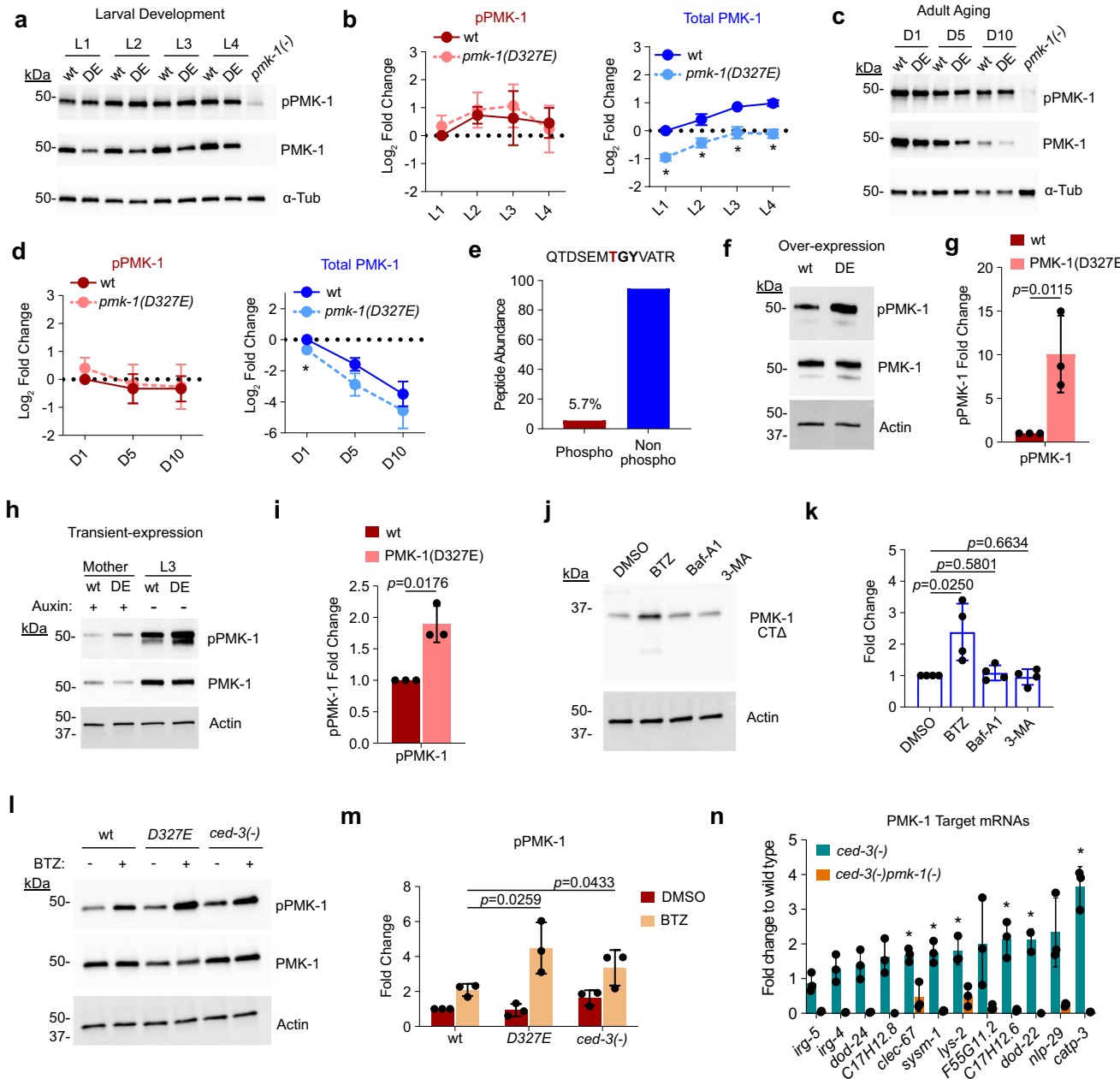

**Fig. 5 | pPMK-1 and non-phospho-PMK-1 expression during development and aging with CED-3 caspase limiting pPMK-1. a–d** Expression of pPMK-1 and total PMK-1. Endogenous *pmk-1* locus tagged with HA with either wild type or cleavage-resistant mutation D327E. *n* = 3 independent biological replicates, relative to wt L1 **b** or Day 1 adult stage **d**. Data are presented as mean fold change relative to wt ± SD. Two-tailed, paired t-test. Asterisk, *p* < 0.05. Exact *p* values provided in Source Data. **e** Mass spec of phospho- and non-phospho-(TGY)-containing peptide abundance. **f-g** Western Blot analysis of pPMK-1 and total PMK-1 from overexpressed wild-type PMK-1 (wt) or PMK-1(D327E) (DE). n = 3 biological replicates, Data are presented as mean fold change relative to wt overexpression ± SD. **h, i** Western Blot analysis and quantification of pPMK-1 and total PMK-1 from transient expression of endogenous wild-type (wt) or cleavage-resistant (DE) PMK-1 controlled by auxin induced degron. *n* = 3 biological replicates. Data are presented as mean fold change relative

to wt ± SD. **j, k** Effects of protein degradation pathways on expression of PMK-1(CTΔ). Western blot analyses following treatment with control (DMSO); Bortezomib (BTZ); Bafilomycin A1 (Baf-A1); and 3-methyladenine (3-MA). *n* = 4 independent experiments. Data are presented as mean fold change relative to DMSO treatment±SD. **l, m** Western blot analysis and quantitation of endogenous pPMK-1 from wt, PMK-1 (D327E) and caspase *ced-3(-)* mutant following Bortezomib treatment. *n* = 3 biological replicates. Data are presented as mean fold change relative to wt DMSO treatment ± SD. **n** Analysis of *pmk-1*-target gene mRNA expression by qRT-PCR in *ced-3(-)* animals. Each bar represents mean fold change relative to wild type. Each dot represent 1 biological replicate. n = 3 biological replicates. Data are presented as mean values±SD. Asterisk, *p* < 0.05. Two-tailed, paired t-test. Exact *p* values provided in Source Data. **b, d, g, i, k, m** *p* values from two-tailed, ratio-paired t-test. Source data are provided as a Source Data file.

obtained when we limited endogenous PMK-1 or PMK-1(D327E) expression using AID inserted by CRISPR mutagenesis at the N-terminus and then allowed transient expression for 1 generation (Fig. 5h, i). These results confirmed that animals tightly regulate the amount of phosphorylated PMK-1. The down-regulation of total

PMK-1(D327E) protein is to compensate for excessive phosphorylated PMK-1. These results also suggest that CED-3 caspase is important in eliminating pPMK-1 because the cleavage resistant *pmk-1(D327E)* mutants have excessive pPMK-1 when expressed transiently.

We then tested whether CED-3 caspase altered accumulation of pPMK-1. We did not observe a cleavage product in vivo using Western blot (Fig. 5a, c) suggesting possibly rapid degradation of cleaved PMK-1. To understand the effect of removing the PMK-1 C-terminus, we used CRISPR mutagenesis to generate an HA-tagged PMK-1 truncated at Asp327 in the endogenous locus *pmk-1(CTΔ)*. We tested 2 lines of the *pmk-1(CTΔ)* mutation and observed only trace levels of the expected ~38 kDa truncated protein (Supplementary Fig. 8b). The lack of PMK-1(CTΔ) expression was not due to reduced mRNA (Supplementary Fig. 8c). To rule out solubility issues, we expressed wild-type PMK-1, PMK-1(D327E) and PMK-1(CTΔ) proteins in vitro using reticulocyte lysates and found all 3 versions of PMK-1 were synthesized to comparable extents in this heterologous system (Supplementary Fig. 8d). Incubating these proteins for 48 h at 20 °C and 37 °C did not result in loss of protein or accumulation of insoluble aggregates (Supplementary Fig. 8d) suggesting no gross differences in solubility and the truncated protein is likely targeted for degradation in vivo.

To identify the clearance pathway that degrades truncated PMK-1, we tested inhibitors of proteasome (Bortezomib, BTZ), lysosome (Bafilomycin A1, Baf-A1), and autophagy (3-methyladenine, 3-MA) and found that blocking the proteasome significantly increased accumulation of truncated PMK-1 (CTΔ, Fig. 5j, k). These findings indicated that the cleaved PMK-1 intermediates are rapidly degraded by the proteasome clearance pathway.

We then tested the effect of proteasome inhibition on accumulation of pPMK-1. We found that animals with either the *pmk-1(D327E)* cleavage-resistant mutation or the *ced-3* caspase null mutation accumulated pPMK-1 by nearly 2-fold more than wild-type animals (Fig. 5l, m and Supplementary Fig. 8e). Again, consistent with our finding that pPMK-1 is maintained at a small fraction of total PMK-1 (Fig. 5e), we did not observe any significant increases in total PMK-1 (Supplementary Fig. 8e). Because *pmk-1(D327E)* mutants have intact CED-3 caspase function and *ced-3* null mutants have wild-type PMK-1, our results suggest that CED-3 caspase limits pPMK-1 in vivo. To further test the effect of CED-3 caspase on PMK-1 signaling, we monitored the changes of PMK-1 downstream target genes in *ced-3(-)* mutant. Consistent with the observation of increased pPMK-1 in *ced-3(-)*, most of the PMK-1 target genes were up-regulated in *ced-3(-)* (Fig. 5n).

## Phospho- to non-phospho-p38 ratio controls growth and aging

Our findings suggested the possibility that altering the phospho- to non-phospho-PMK-1 ratio may impact net PMK-1 signaling. There is no suitable phospho-mimetic mutation for the TGY dual phospho-site that is able to mimic pTGpY. However, VHP-1 is a dual specificity phosphatase (DUSP) required to dephosphorylate PMK-1[5,6]. Therefore, we used *vhp-1(RNAi)* to increase accumulation of phosphorylated PMK-1.

Losing *vhp-1* function during larval development by RNAi causes a developmental delay for wild-type animals (Supplementary Fig. 9a) and this phenotype is *pmk-1*-dependent as seen by alleviation of stalled development in the absence of functional *pmk-1* (Supplementary Fig. 9a). The stalled development we observe in this system is consistent with previous findings by us and others when PMK-1 signaling is hyperactivated[5,29–31]. This treatment is therefore a useful tool to enhance phosphorylated PMK-1 levels to study PMK-1 signaling. To test the relationship of phospho-PMK-1 ratio to non-phospho-PMK-1 further, we examined rates of growth when altering both pPMK-1 and the total pool of PMK-1. For additional PMK-1 expression, we used a transgene with constitutive promoter (*eft-3*) and permissive 3′ UTR element (*unc-54* 3′ UTR) to overexpress PMK-1 controlled by auxin-induced degron.

We observed that animals fed *vhp-1(RNAi)* without PMK-1 transgene expression (auxin treated) had a marked developmental delay compared to mock (RNAi) (Light green, Fig. 6a). Interestingly, animals fed *vhp-1(RNAi)* but allowed to overexpress PMK-1 protein by the

transgene were significantly protected against the developmental delay with nearly half the animals reaching adulthood (Dark green, Fig. 6a). When we examined the accumulation of pPMK-1 and total PMK-1, we find that *vhp-1* RNAi-treated animals all had similar absolute amounts of pPMK-1 (Fig. 6b). However, the animals showing the strongest developmental delay also had the highest pPMK-1 to PMK-1 ratio (Fig. 6c and Supplementary Fig. 9b). Moreover, we observed a similar result of overcoming the *vhp-1(RNAi)* delay by overexpressing a *pmk-1* mutant that cannot be phosphorylated (TGY mutated to AGF) (Supplementary Fig. 9c-d). These results suggest that increasing non-phosphorylated PMK-1 pool mitigated the effect of hyperactivated pPMK-1.

The observation that animals adapt to maintain the absolute amount of phosphorylated PMK-1 with caspase cleavage resistant mutation *pmk-1(D327E)* prompted us to test whether this tight regulation of pPMK-1 also exists when overexpressing wild-type PMK-1. When allowing animals to overexpress wild-type PMK-1 transgene, we observed that the total PMK-1 protein increased by about 40-fold but strikingly, pPMK-1 was unchanged compared to the endogenous level (Fig. 6d). This result confirms that animals strictly regulate the absolute amount of pPMK-1 in the absence of stress.

In the case of *pmk-1(D327E)* mutants, signaling increased as the total PMK-1 decreased. We then tested the impact on signaling when total PMK-1 increases. Because it is technically unfeasible to maintain the PMK-1 overexpression as animals age, we tested overexpression of PMK-1 with developmental phenotypes. We previously showed that loss of *pmk-1* function results in faster larval development[29]. Consistent with this finding, knocking down endogenous PMK-1 protein with the auxin-induced degron (AID) system resulted in faster growth (Endo -, Fig. 6e). Interestingly, we also found that mutants with overexpressed PMK-1 phenocopied loss of *pmk-1* function with faster growth rate (O/E +, Fig. 6e). Moreover, when we overexpressed PMK-1, lysosome particle size was enlarged similar to *pmk-1* null (Fig. 6f).

These results were counterintuitive because for most proteins, overexpression results in enhanced function. However, the observation that altering the non-phosphorylated PMK-1 pool inversely impacts PMK-1 function underscores the importance of stoichiometry in modulating signaling (Fig. 6g). In the case of the *pmk-1(D327E)* mutation, signaling is enhanced when non-phospho-PMK-1 is decreased. Conversely, in the case of overexpression, signaling is diminished when non-phospho-PMK-1 is increased (Fig. 6g). In both cases, absolute pPMK-1 amount is not changed. Therefore, alteration of non-phospho-PMK-1 provides an additional mechanism for organisms to modulate PMK-1 signaling distinct from stress induced hyperactivation of PMK-1.

We observed in wild type animals, the amount of pPMK-1 is fairly constant during early aging but non-phosphorylated PMK-1 is dramatically down regulated (Fig. 5c, d). This natural phenomenon gave us the opportunity to test the impact of pPMK-1 to non-phospho-PMK-1 stoichiometry on signaling. Based on the Western blot quantification (Fig. 5b, d) in combination with mass-spectrometry (Fig. 5e), we estimated the ratio of pPMK-1 to non-phospho-PMK-1 increased from 1:17 on Day 1 to about 1:1 on Day10 (Fig. 6h). This ratio is further enhanced in *pmk-1(D327E)* mutants (Fig. 6h). We then tested expression of a dozen *pmk-1* downstream target genes on Day 1 and Day 5 of aging. Expression of these genes were increased from 2 to 100 fold on Day 5 compared to Day 1, suggesting enhanced PMK-1 signaling in wild-type animals (Fig. 6i). During this period of aging, pPMK-1 amount was unchanged and total PMK-1 was diminished (Fig. 5c, d). Therefore, the increased signaling paralleled the increased pPMK-1 to non-phospho-PMK1 ratio.

We further confirmed this observation using the well-established downstream PMK-1 target reporter *P_sysm-1_*::GFP. We found that consistent with gene expression results, wild-type animals increased expression of *P_sysm-1_*::GFP by day 5 compared to day 1 adulthood

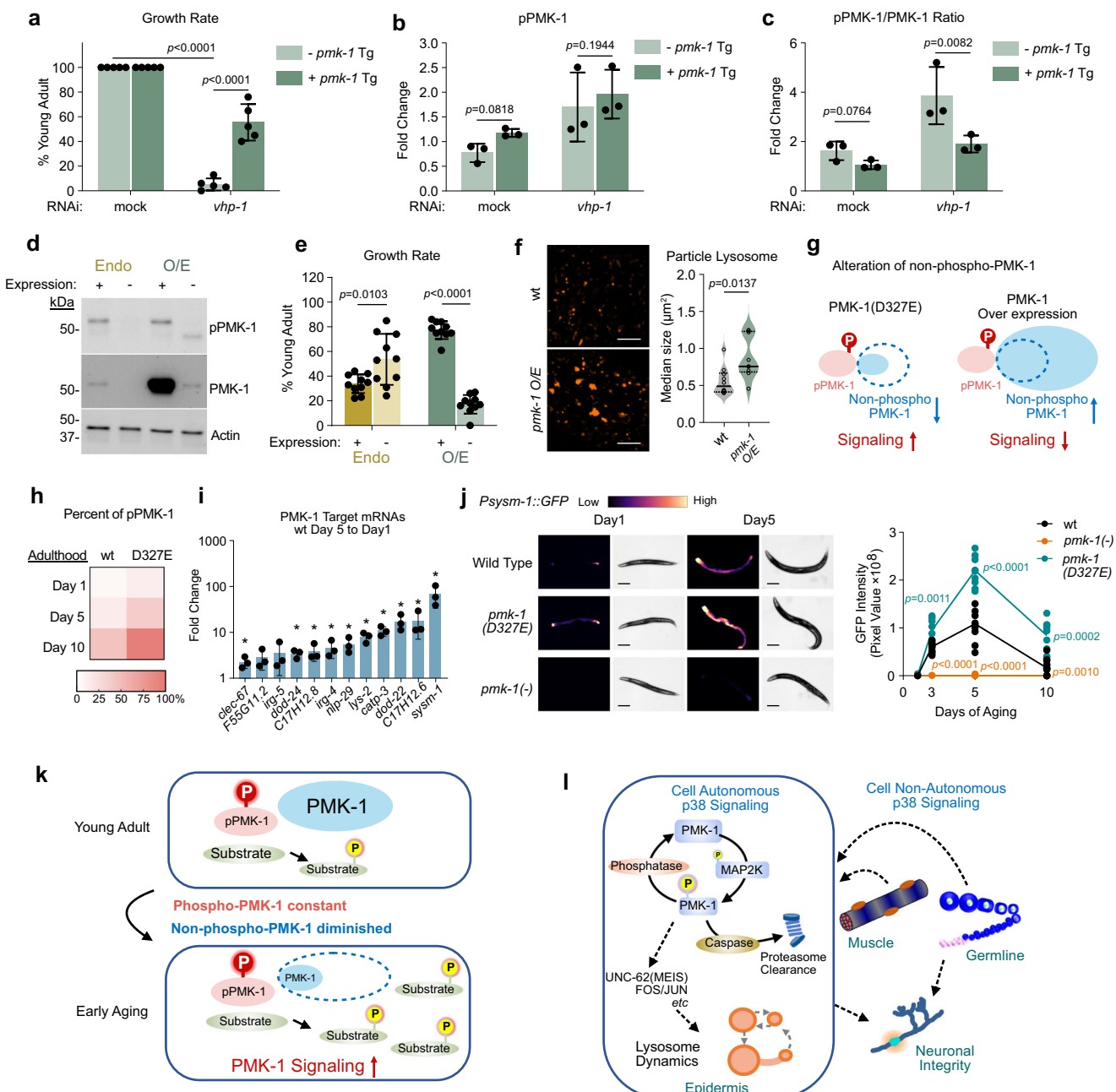

**Fig. 6 | Altering the ratio of phospho-PMK-1 to non-phospho-PMK-1 regulates growth and longevity. a** Rescue effect of expressing PMK-1 transgene (Tg) on *vhp-1* RNAi-induced developmental delay. Each dot represents one plate with 20–40 animals per condition. *n* = 5 biological replicates. Data are presented as mean value ±SD. *p* value from two-tailed, unpaired t-test. **b** Quantification of Western blot showing similar pPMK-1 accumulation with or without PMK-1 overexpression. **c** Altering pPMK-1 to total PMK-1 ratio using *vhp-1* RNAi and PMK-1 transgene. **b**, **c** each dot represents one biological replicate, data are presented as mean value ±SD. *n* = 3 biological replicates. *p* value from two-tailed ratio-paired, t-test. **d** Western blot of HA-AID-tagged PMK-1 from endogenous locus (Endo) or transgene overexpression (O/E). **e** Growth rate assays for HA-AID-tagged PMK-1 from endogenous locus (Endo) or overexpression (O/E). Synchronous larvae developed for 60 h. Each dot represents one plate of 25-40 animals. *n* = 10 biological replicates. Data are presented as mean value ± SD. *p* value from two-tailed, unpaired t-test. **f** Day 1 adult epidermal particle lysosome (P*CED-1*::NUC-1::mCherry). Scale bar,

10μm. Violin plot, each dot represents median values of all lysosome particles imaged per animal. wt n = 10, O/E n = 9 animals. *p* value from two-tailed, unpaired Welch's t-test. **g** Model showing signaling outcome of altering non-phospho-PMK-1. Dashed circle indicates wild type non-phospho-PMK-1 quantity. **h** Estimated percent of phospho-PMK-1 in early aging using Western Blot quantification n = 3 (Fig. 5c, d) and Mass-Spec (Fig. 5e). **i** Analysis of *pmk-1*-target mRNAs by qRT-PCR during early aging. Each bar represents mean fold change±SD of Day 5 relative to Day 1. Each dot represents 1 biological replicate. n = 3 biological replicates. Asterisk, *p* < 0.05. Two-tailed, paired t-test. Exact *p* values provided in Source data. **j** The *pmk-1*-dependent P*sym-1*::GFP reporter expression. Scale bar, 200 μm. Each dot represents total GFP intensity of 1 animal, *n* = 10 animals for each genotype. *p* value from two-tailed, unpaired t-test. **k** Model showing the non-phospho-PMK-1 pool impacts PMK-1 signaling during early aging. **l** Model showing both cell and non-cell autonomous PMK-1(p38) signaling supporting tissue homeostasis during aging. Source data are provided as a Source Data file.

(Fig. 6j). Furthermore, the *pmk-1(D327E)* mutant had more enhanced $P_{sysm-1}$::GFP signal through day 5 of adulthood (Fig. 6j and Supplementary Fig. 9e). The increased signaling of *pmk-1(D327E)* mutants is the result of those mutants having a higher ratio of pPMK-1 to non-phospho-PMK-1 than wild-type animals (Fig. 6h). We further observed that the $P_{sysm-1}$::GFP reporter showed a diminished signal by day 10 adulthood compared to day 5. This may be due to either that total PMK-1 pool is reduced beyond a threshold required to sustain signaling or the impact of global alterations in translation as animals age. Altogether, our results demonstrate that during both developmental growth and early aging, the net PMK-1 signaling output is modulated by altering the amount of non-phosphorylated PMK-1 (Fig. 6k).

In total, our findings support a model whereby in addition to phosphorylation status, proteostasis mechanisms modulate PMK-1 function by regulating the balance of pPMK-1 and total PMK-1 pools. Moreover, PMK-1 has both cell autonomous and cell non-autonomous functions in supporting homeostasis by promoting lysosome function, an important clearance pathway for proteostasis, as well as neuronal integrity supporting longevity (Fig. 6l).

## Discussion

Maintenance of gene expression programs, proteostasis and tissue integrity during growth and aging requires coordinated signaling across tissues[25]. Although global transcriptional programs are dramatically altered during aging in both worms and mice[56,57], recent findings suggest that aging tissues remain responsive to both positive and negative cell non-autonomous cues. In mice, declines in the transcriptome within neurogenic regions of the brain can be rejuvenated by either physical exercise or humoral factors from young animals[57]. In *C. elegans*, negative cell non-autonomous germline-to-soma Hedgehog signaling in mated hermaphrodites leads to tissue diminution[58]. Cell non-autonomous signaling was also found in flies to support tissue homeostasis including dFOXO signaling[59], rescue of muscle mitochondrial cristae formation by expressing PINK1 in neurons[60,61] as well as BNIP3-mediated mitophagy in neurons supporting homeostasis of muscle and intestine[62].

Recent advances in AID-protein tagging and tissue-specific expression of TIR1 E3 ligase in *C. elegans* have provided powerful tools to analyze cell autonomy[46,47]. In this study, we find that germline and epidermal p38 MAPK functions are both critical to support pain sensory neuron integrity and function during early aging in *C. elegans* (Fig. 6l). Further, germline and muscle p38 MAPK functions were required to support epidermal lysosome formation by early adulthood (Fig. 6l). As additional high-fidelity tissue-specific technologies are developed, future studies can further refine the identity of the signaling cell. Despite not understanding the intercellular signaling molecules in many cases, it is clear that signaling is integrated between tissues to support longevity and remains an exciting area for further study.

As a further complexity of cross-tissue regulation, previous work showed that the *pmk-1(-)* null mutant animals treated with FUDR to eliminate germline proliferation had normal life span in the absence of infection[9,12]. Here we show that the germline p38 MAPK function is critical to support neuronal integrity and epidermal lysosomal structures to promote longevity when germline is actively proliferating. Further, it was recently shown that PINK-1 promotes mitophagy to support longevity of the reproductive span for *daf-2* mutants in *C. elegans*[63]. Altogether these results are consistent with the idea that the proteostasis load of germline function has a complex interplay in longevity[28,32].

From our findings, we raised the question how altering the relative pools of phospho- to non-phospho-PMK-1 could modulate or tune signaling. Previous work has also implicated an antagonistic role of mono-phosphorylated Fus3 MAPK to dual-phosphorylated Fus3 and the dynamic role of the membrane-associated scaffold protein Ste5 in response to yeast mating pheromone[64]. A previous model predicted

that MAPK interactions with scaffold proteins may ultimately regulate net signaling output from MAPKs[65]. The idea of signaling optima is consistent with the observation that modulating the pPMK-1 to non-phospho-PMK-1 ratio altered signal strength despite no change in the absolute amount of pPMK-1.

In this study, we show that wild-type animals exhibit enhanced p38 signaling in early aging by reducing the total PMK-1 pool while keeping the pPMK-1 pool constant (Fig. 6k). By maintaining the absolute amount of pPMK-1, p38 signaling is enhanced for the maintenance of tissue homeostasis without evoking hyper-activation of stress-responses. However, the tradeoff is a smaller total p38 reservoir for immune function in the case of stress response, consistent with previous findings that loss of *pmk-1* function is responsible for immunosenescence in aging[13].

In addition to responding to diverse environmental stimuli, p38 MAPK signaling regulates cellular homeostasis and growth. Previous findings have described a complex relationship of p38-dependent innate immune responses and neuronal degeneration[66–68] as well as modulating adaptations to stressors like hypoxia including glutamate receptor trafficking in aging neurons[69].

Physiological coordination of stress responses and proteostasis requires both cell autonomous and non-cell autonomous cross-tissue signaling. Within neurons, HSF-1 was shown to have a cell autonomous effect on neuronal integrity which was correlated with retention of locomotory function during aging[52]. The olfactory neuronal gene *olrn-1* suppresses the p38 MAPK PMK-1 immune pathway in the intestine[30] Additionally, a tradeoff between immunity and longevity was shown to be mediated by the transcription factor PITX1/UNC-30 with non-cell autonomous signaling from GABAergic neurons to limit intestine PMK-1 activation[70]. Previous work established a key role of the nervous system in regulating stress responses in other tissues including the UPR in intestine[71]. More recent studies have illuminated that MAPK-dependent regulation of hyaluronidase (TMEM2) promotes ER homeostasis[72] and degradation of the p38 target MK2 functions as a feedback signal critical for stress-induced cell fate[73] as well as emerging evidence that p38 signaling has a crucial role in synaptic plasticity and neurodegenerative processes[74]. Our previous findings revealed that PMK-1 coordinates growth rate with stress responsiveness during larval development. Here we show that PMK-1 promotes tissue integrity in the absence of stress by modulating the extent of non-phosphorylated PMK-1. Altogether, how p38 coordinates diverse downstream targets and the mechanisms of tuning MAPK signaling to optimize competing physiological demands remains an active area of research.

Lysosomal clearance has a complex relationship in neuronal homeostasis. Defective biogenesis and repair of lysosomes are hallmark features of neurodegenerative pathologies[42]. In contrast, other work has shown that lysosomes are required for necrotic cell death-mediated neurodegeneration and disruption of lysosomal biogenesis and fusion can be protective to necrosis[75]. Lysosomes have recently been identified in a fat-to-neuron lipid-signaling function in *C. elegans* activating neuropeptide signaling to promote longevity[76]. Recent findings have also revealed that preserving germline integrity requires the nucleophagic clearance pathway[77]. Our current study unmasked p38 MAPK signaling from multiple tissues including germline to support a transcriptional network to maintain epidermal lysosome function. Moreover, PMK-1 signaling in epidermis supports sensory neuron integrity (Fig. 6l). Altogether, these findings underscore the importance of clearance pathways for proteostasis in tissue homeostasis and longevity.

Following injury, CED-3 caspase was previously shown to promote neuronal regeneration in *C. elegans*[78] suggesting a caspase may have multiple targets in neuronal protection. Further, the observation that loss of CED-3 caspase did not always correspond to cleavage-resistant PMK-1 phenotypes underscores the complexity of the cross-talk

between the caspase and MAPK signaling pathways. Understanding non-apoptotic caspase functions is hindered by challenges in monitoring the dynamic functions at low levels of caspase activation. Future studies will certainly prove out the intricacies of caspases working to sculpt multiple aspects of signal transduction in these and other functions.

Altogether, we show that the enhanced p38 signaling during early aging is achieved by lowering the stoichiometry of non-phosphorylated p38 protein without affecting the amount of phosphorylated-p38 (Fig. 6k). In this scenario, the enhancement of p38 signaling during early aging is protective to tissue homeostasis likely through enhanced gene expression and protein clearance mechanisms supporting proteostasis (Fig. 6l). Beyond natural tuning of the p38 phospho-ratio during normal aging, whether this can be exploited for therapeutic purposes is an intriguing open-ended question.

## Methods

### Protein modeling
AlphaFold[79] was used to generate a model of *C. elegans* PMK-1 using p38α (PDB code 2OKR) and p38γ (PDB code 6UNA). Model confidence around the caspase cleavage site is very high (pLDDT > 90).

### Strains used in this study and culturing conditions
Strains used in this study, Supplementary Data 5. *C. elegans* culture conditions and RNAi treatment as previously described[29]. NGM agar plates with OP50 lawn were used for standard maintenance, growth and aging assays as indicated. For RNAi experiments, HT115 culture was seeded on NGM agar plates containing 1 mM IPTG and 200 μg/mL ampicillin for indicated RNAi treatments and assays. For all assays, prior to testing, animals were maintained under stress-free conditions at 20 °C for multiple generations, including no starvation and no obvious contaminations.

### CRISPR-Cas9 mutagenesis
For list of all mutants generated by CRISPR-Cas9 mutagenesis used in this study, see Supplementary Data 5. Endogenous HA-AID-GS and HA-GS-tagged *pmk-1* or HA-GS-tagged *pmk-1(D327E)* mutant and HA-AID-GS-tagged *pmk-1(D327E)* were generated by CRISPR/Cas9. In brief the rescue DNA was assembled using NEB HiFi DNA assemble Kit and purified PCR product was used as the rescue template. The AID degron tag was amplified from pLZ31[47] with modification. The injection constructs mix containing the sgRNA target the 1st exon of *pmk-1* (pDD162_C0025) were injected into the N2 strain or the *pmk-1(D327E)* mutant[29] respectively. To generate the overexpression mutant, a single copy of *ha-aid-gs-pmk-1, ha-aid-gs-pmk-1(D327E)* or *ha-aid-gs-pmk-1(T191A, Y193F)* flanked by the *eft-3* promoter and *unc-54* 3′ UTR were inserted into Mos I site oxTi365 on Chr V by CRISPR/Cas9 following the SEC methods[80]. The overexpression mutants were generated by injecting constructs into the CA1200 strain[47]. For tissue-specific TIR1 expression, we used promoters previously established with TIR1 E3 ligase[46,47].

### Auxin-induced degron (AID) system culture conditions
To conditionally inactivate PMK-1 or over-express PMK-1, we used the Auxin-Induced Degron (AID) system. The auxin analog K-NAA was used in lieu of auxin for this study. For culture conditions, we sterile-filtered the auxin analog K-NAA dissolved in MilliQ $H_2O$ and added to standard NGM agar at a final concentration of 0.1 mM K-NAA followed by seeding of OP50 as normal. We find that adults maintained on K-NAA plates transferred to standard NGM/OP50 media begin to restore PMK-1 expression within 24 h and reach wild-type levels within another 2 days of adulthood.

For tissue-specific PMK-1 degradation, endogenous PMK-1 was tagged with HA-AID at the N-terminus using CRSIPR-Cas9 mutagenesis

and crossed into strains carrying TIR1 E3 ligase expressed under different promoters as previously described[46,47] including: whole body (*eft-3*, CA1200), Intestine (*ges-1*, CA1209) and germ line (*sun-1*, CA1199) as well as additional strains generated in this study using SEC insertion method in LGII ttTi5605 MosI site including: epidermis (*col-10*), body wall muscle (*myo-3*), pan-neuron (*rgef-1*). For cell autonomy tests, the no auxin treatment was done with water control on plates seeded with TIR1 RNAi in HT115 to limit background degradation and auxin treatment was done with 0.1 mM auxin in the agar and plates were seeded with mock RNAi in HT115.

### Growth rate assay
Five gravid young adults were allowed to lay eggs for 1 h on each plate and removed. Animals were staged at the 60 h post egg laying for percentage reach adulthood.

### Aging assay
Mid-L4 stage animals were placed on OP50 NGM (without FUDR) plates with 20 animals per plate and 5 plates per genotype. The next day animals were confirmed to be egg-laying and this was defined as Day 1 of adulthood. Worms were moved to a fresh NGM plates daily for the first 5 days and subsequently every 4-5 days.

### Imaging
The epidermal lysosomal marker ($P_{CED-1}$::*nuc-1*::mCherry)[43], the pan-neuronal marker ($P_{RGEF-1}$::DsRed)[54], and the PVD/AQR-neuronal marker ($P_{F49H12.4}$::GFP from NC1687) were imaged after worms were mounted on Noble agar pads in M9 with $NaN_3$ (0.25%). Zeiss AxioImager M2 with a black and white Hamamatsu ORCA C13440 camera was used for DIC images. All images were taken at the same magnification for the same amount of time for comparisons as indicated in the relevant figure legends. Fiji (Image J) was used for lysosome quantification. The same threshold was applied to all mutants. For lysosomal structure, puncta with circularity of 0.4 to 1 were quantified as particles and puncta with circularity of 0–0.4 were quantified as tubules.

### Harsh touch behavioral assay
Adults were cultured on OP50 NGM plates (without FUDR) and were tested on indicated days of adulthood prior to transferring. Adults were transferred daily. To test, animals were lightly poked with a standard platinum wire pick in the anterior segment just behind the pharynx. Animals that reversed course immediately were scored as "correct response". Animals were allowed up to 5 s before scoring a response. A total of 20 animals per genotype were tested per day per run. 3 independent runs were performed for a total of 60 animals per genotype.

### Western blots
Synchronous L1 stage animals were obtained following standard alkaline bleach to obtain eggs as previously described[17]. L2, L3, L4, and day 1 adults were collected at 8, 18, 30, and 48 h post-feeding. For adults, 300 animals per sample were manually transferred daily and indicated days of adulthood were harvested. Animals were washed off plates, washed 4 times with M9, and pellets were snap frozen with liquid nitrogen. Pellets were sonicated in lysis buffer containing 10 mM Tris pH 7.4, 1 mM EDTA, 150 mM NaCl, 0.5% NP-40, Halt™ Protease and Phosphatase Inhibitor Cocktails (Fisher Scientific, PI78440) and kept cool using ice. BCA assay was used to determine protein concentration. Approximately 4–6 μg of total protein was loaded per well and resolved on 4–20% gradient acrylamide SDS gels (BioRad, 4568096).

To delineate the extent of transcription factor phosphorylation, strains bearing GFP fusions to *unc-62, fos-1*, and *jun-1*[48,49] were treated with mock or *pmk-1* RNAi. Lysates were separated and analyzed using 4–20% gradient acrylamide SDS gels and SuperSep™ Phos-tag™ 7.5% gels (Wako Chemicals 198-17981) to monitor phosphorylation of transcription factors. The SuperSep™ Phos-Tag™ gel delays the

phosphorylated form of the protein during gel electrophoresis. Isoforms were estimated based on molecular weight. The phosphorylated forms were determined by comparison of SuperSep™ Phos-tag™ and SDS gels by probing with anti-GFP antibodies.

Anti-HA antibody from Rabbit (Cell Signaling Technology, 3724 S) or from Mouse (Cell Signaling Technology, 2367 S) were used at 1:1000 dilution. Anti Phospho-p38 (Cell Signaling Technology, 4511 S) was used at 1:2000 dilution. Anti-mCherry antibody (Cell Signaling Technology, 43590) and anti-GFP antibody (Cell Signaling Technology, 2956) were both used at 1:1000 dilution to detect fluorescent fusion proteins. GFP Magnetic Bead Conjugate (Cell Signaling Technology, 67090) was used for IP of GFP fusion protein at 1:100 dilution in lysate. Secondary antibodies were anti-mouse IgG HRP (Cell Signaling Technology, 7076) and anti-rabbit IgG HRP (Cell Signaling Technology 7074) both used at 1:10,000 dilution. Anti-α-tubulin (Sigma-Aldrich, T5168) or Anti-Actin (Bio-Rad, 12004163) antibodies were used at 1:4000 and 1:2000 respectively for loading controls as indicated. For uncropped Western blots, see Source Data file.

### In vitro stability assay
TNT Rabbit reticulocytes were used to radiolabel proteins with $^{35}$S-Methionine for 1 h at 30 °C. Products were then diluted 1:5 into 25 mM Tris, pH 8.0, 0.25 mM EDTA, 0.25 mM sucrose, and 2.5% glycerol. Mock reactions were collected directly into 3 volumes of 2× SDS buffer with 5% β-mercaptoethanol. Two additional reactions were incubated at either 20 °C or 37 °C for 48 h and then quenched with 3 volumes of 2× SDS buffer. One-fourth of samples were resolved on 4–20% acrylamide gels, fixed with 50% methanol and 10% glacial acetic acid, protected with 10% glycerol, and dried onto Whatman filter paper. Dried gels were exposed to phospho-imager screens over 2 nights and imaged on phospho-imager.

### Proteasome, lysosome and autophagy inhibition assays
Synchronous L1 stage animals were fed OP50 for 40hrs and washed off plates into S-basal containing Bortezomib (130 µM), Bafliomycin-A1 (50 µM), 3-MA(50 mM) or DMSO (0.1%) with OP50 food. The liquid culture was incubated at 20 °C for 6 h with 250 rpm shaking. Then the worms were washed 4 times with M9 and the pellet was frozen in liquid N2 and stored at -80 °C until Western Blot analysis.

### mRNA-Seq and transcription factor binding site analyses
Synchronous young adult stage animals were collected into TRIzol™ Reagent (Invitrogen, 15596-026). Total RNA was extracted. Three replicates each of wild-type, pmk-1(-), and pmk-1(D327E) mutants were collected. For daf-16 mutants, daf-16(-);pmk-1(-) and daf-16(D327E) mutants, 2 replicates each for m26 and mgDf50 alleles were used. Libraries were generated, sequenced and analyzed for differential gene expression by the UTSW McDermott Center Next Generation sequencing core. Samples were run on the Agilent Tapestation 4200 to determine extent of degradation thus ensuring only high-quality RNA was used (RIN Score 8 or higher). The Qubit fluorimeter was used to determine the concentration prior to starting library preperation. One microgram of total DNAse-treated RNA was then prepared with the TruSeq Stranded mRNA Library Preperation Kit from Illumina. Poly-A RNA was purified and fragmented before strand specific cDNA synthesis. cDNA were then a-tailed and indexed adapters were ligated. After adapter ligation, samples were PCR amplified and purified with AmpureXP beads, then validated again on the Agilent Tapestation 4200. Before being normalized and pooled, samples were quantified by Qubit then run on the Illumina NextSeq 500 using V2.5 reagents. Wormbase gene set enrichment analysis tools were used for gene ontology analyses. Transcription factor binding sites analysis was done using modENCODE (from Wormbase JBrowse) database. All the TFs binding to ~2 kb upstream of the transcription start site were recorded and data from all developmental stages were pooled (Supplementary Data 3).

### Reverse transcription and quantitative PCR
Total RNA was extracted using TRIzol™ Reagent (Invitrogen, 15596-026) from animals with different genotypes and developmental stages. For cDNA synthesis, 1 µg of total RNA was used as input for all samples. The iScript RT Supermix for RT-qPCR (BioRad, 1708841) was used for cDNA synthesis. The CFX Realtime PCR system (BioRad) with iTaq Univeral SYBR Green Supermix (BioRad, 1725122) was used for qPCR analysis. Each condition was tested with 3 biological replicates. C. elegans ama-1 was used as the reference gene for normalization in all experiments. Primers used for qRT-PCR analysis (Supplementary Data 6).

### PMK-1 p38 MAPK PTM quantification and mass spectrometry
Synchronized young adult animals expressing N-terminal HA-tagged PMK-1 were used for immunoprecipitation with anti-HA-Tag (C29F4) rabbit mAb (Cell Signaling Technology 11846) and eluted with 2× laemmli buffer. Eluted protein was resolved by SDS-PAGE and gel slices of PMK-1 band were sent for protein ID using Mass Spectrometry. Trypsinization, desalting, mass-spectrometry and analyses were performed by the UTSW proteomics core. For phospho-peptide analysis, samples were run on an Orbitrap Fusion Lumos mass spectrometer coupled to an Ultimate 3000 RSLC-Nano liquid chromatography system. Data analysis was performed using Proteome Discoverer 2.4 SP1 using the C. elegans protein database from UniProt. The ratio of the abundance of T196 phospho-peptide to the unphosphorylated peptide was used to estimate the percentage of phosphorylated PMK-1.

### Estimating relative pPMK-1 fraction through aging
Generating sufficient quantity of aged C. elegans samples for mass-spectrometry is limiting. Thus, we estimated the relative fraction of pPMK-1 based on mass-spectrometry measurements of day 1 adults. Western blots of 3 independent biological samples from wild type and pmk-1(D327E) mutant animals for day 1, day 5, and day 10 were quantified for pPMK-1 and total PMK-1 with Image Lab (Bio-Rad) and normalized to actin. Mean values of 3 experiments were used to determine the relative fold change of protein accumulation between different genotype and throughout days of aging (relative to Day 1 wild type). Day 1 wild type pPMK-1 was then set to 5.7% based on mass-spectrometry measurements and estimates of pPMK-1 fraction of all other conditions were calculated based on Western Blot quantification.

### Lys48 linkage ubiquitin pull down and mass spectrometry
Synchronized young adult animals were treated with Bortezomib (60 µM) for 8 h and worm pellets were washed 4 times and flash frozen in liquid nitrogen. Polyubiquitinated proteins were enriched using Ubiquitin Enrichment Kit (Thermo Scientific 89899) and eluted with 2× laemmli buffer. Lys48-specific anti-Ubiquitin rabbit clone Apu2 (Sigma 05-1207) was used for immunoprecipitation at 4 °C for 4 h. The immune-complex was captured using Pierce MS-Compatible Magnetic IP Kit Protein A/G (ThermoFisher 90409) and eluted with 2× laemmli buffer. Eluted proteins from both enrichment methods were resolved by SDS-PAGE and gel slices above 50 KDa were sent for protein ID using mass spectrometry. Ubiquitin antibody (P4D1, Cell Signaling Technology, 3936) was used for western blot to detect ubiquitinated proteins at 1:1000 dilution. Mass-spec and protein identification were performed by the UTSW proteomics core. For protein identification, samples were run on a Q-Exactive HF mass spectrometer coupled to an Ultimate 3000 RSLC-Nano liquid chromatography system. Data analysis was performed using Proteome Discoverer 3.0 SP1 using the C. elegans protein database from UniProt.

### Statistics and reproducibility
Statistics were reported in the Legends with samples sizes and p values indicated throughout. Replicates and specific p values are provided in supplementary figures and source data.

**Reporting summary**

Further information on research design is available in the Nature Portfolio Reporting Summary linked to this article.

## Data availability

The mRNA-Seq data generated in this study have been deposited in the GEO database under accession code GSE192941. The remaining data generated in this study are provided in the Source Data file. Source data are provided with this paper.

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

## Acknowledgements

We thank David Mangelsdorf, Leon Avery, Duojia Pan, James Collins, Michael Reese, Matthew Sieber and David Corey for helpful discussions, the CGC (funded by NIH Office of Research Infrastructure Programs [P40 OD010440]) for materials; WormBase and UniProt databases. mRNA-seq was performed by the McDermott Center Sequencing Core and data analysis was provided by the McDermott Center Bioinformatics Lab. Protein mass-spectrometry was performed and analyzed by the UT Southwestern Proteomics Core facility. This work is supported by The Robert A. Welch Foundation grants I-2022-20190330 (BPW) and I1243 (MHC), and National Institutes of Health grant R35GM133755 (BPW). The funders had no role in study design, data acquisition, decision to publish, or preparation of the manuscript.

## Author contributions

W.Y., Y.M.W., M.H.C., and B.P.W. conceived the study and designed research; W.Y., Y.M.W., S.E., C.A.T., and B.P.W. performed research; W.Y., Y.M.W., S.E., C.A.T., M.H.C., and B.P.W. analyzed data; W.Y., Y.M.W., M.H.C., and B.P.W. wrote the manuscript; Y.M.W., M.H.C., and B.P.W. supervised the study; M.H.C. and B.P.W. acquired funding.

## Competing interests

The authors declare no competing interests.
