## [Peer Review File · Nature Communications]

Modulating p38 MAPK Signaling by Proteostasis Mechanisms Supports Tissue Integrity during Growth and AgingREVIEWER COMMENTS

Reviewer #1 (Remarks to the Author):

Yuan et al. characterize the consequences of altered PMK-1-p38 signaling in *C. elegans*. They use a mutant in the previously identified CED-3 cleavage site and a loss of function mutation. They report that loss of pmk-1 reduces lifespan, contradicting a previous study that showed that in daf-2 but not WT worms lifespan is significantly reduced by a pmk-1 mutation. In contrast, the cleavage resistant mutant showed extended lifespan. Transcriptome experiments showed a differential impact of loss of function and cleavage resistant mutations in pmk-1. Based on these results they characterize lysosomes and subsequently neuronal integrity. Finally, they investigate the ratios of phosphorylated and total PMK-1 depending on the cleavage site.

The role of PMK-1 stress signaling in aging is generally interesting as it pertains to stress and immune signaling. The current manuscript suffers from lack of essential controls and the unstructured organization of the manuscript makes it very confusing and hard to follow.

Major concerns:

1. The degron strains are not sufficiently characterized. There are no controls for the degree of degradation of PMK-1 and, importantly, there is no control of the tissue specificity of the PMK-1 inactivation. This is an essential point because PMK-1 was previously shown to predominantly function in the intestine but endogenous PMK-1 protein has not been detected in any other cell type. This must be convincingly shown to support any of the non-cell autonomous claims of the paper. The degron strains must be carefully characterized before they are used as evidence. Moreover, no mechanism for non-cell autonomous roles are being explored. These findings are purely correlative.
2. It is important to show the consequences of the cleavage site mutant on pPMK-1/PMK-1 levels and the consequence on PMK-1 activity alongside the lifespan experiments in Figure 1. They are now shown at the end and for example Figure 6h is unclear. This panel is essential to substantiate the claims of the paper that the ratios of pPMK-1/PMK-1 play an important role in the activity of PMK-1. It appears that this heatmap is a conundrum of different data types and the original data need to be shown and carefully quantified.
3. Figures 2F, G are unclear. It appears there were two replicates that differ and this requires further repeats to solidify the data. The link to PMK-1 are unclear and the findings seems unconnected. The role of PMK-1 signaling in protein homeostasis remains unclear.
4. How the lysosomes are regulated by the pPMK-1/PMK1 ratio remains mechanistically unclear. What is the role of the lysosomal aberrations in the consequence of pmk-1 loss of function?
5. The link between PMK-1 and the transcription factors and lysosomal genes in Figure 3f are unclear. For example, is the activity of UNC-62, FOS-1 and JUN-1 in pmk-1 null or degron mutants affected?
6. The connection between lifespan, lysosomes and pain sensory neurons is unclear.
7. The role of the cleavage site in vivo remains rather unclear based on the results presented in Figure 5. For instance epistasis experiments in Figure 1 with ced-3 and other outcomes of PMK-1 signaling would help clarifying this.
8. Numbers of experimental repeats etc. are lacking throughout.

Additional concerns:

1. 128-130: The cleaved protein is not functional in the context of lifespan. Can it have other functions?
2. 143-144: They claim that cleavage resistant pmk-1 could not restore daf-16 mutants to normal lifespan. There is no wt control neither in figure 1f nor in Supplementary 1d. Was this lifespan performed in parallel to the one in 1e?
3. 162-164 "Most genes altered by loss of pmk-1 are also affected by loss of daf-16, suggesting PMK-1 signaling and DAF-16 co-regulate these genes": reference? Is this known also for genes other than the ones involved in innate immune response?
4. Supplementary Fig 2: In a and b are the heatmaps compared to wt? Expand legends.
5. Fig. 2f: controls are missing.

6. Fig. 3a: how are the tubules and particulates defined?
 7. Fig. 3c: If the lysosome enlargement is happening from L1 to L4, is it a developmental phenotype or an ageing phenotype?
 8. Fig 3e: Why do neurons and intestine have no effect?
 9. Fig. 3f: Can it be that the transcription factors KD is affecting particle size independently of pmk-1?
- 258-259: The suggestions that PMK-1 signals through a complex transcriptional network to regulate lysosomal genes is not well supported.

Reviewer #2 (Remarks to the Author):

The paper provides interesting knowledge about the complex function of PMK-1 enzyme family in *C. elegans* tissue homeostasis and longevity, in particular describing the critical role of non-phospho-p38 stoichiometry in the proper balance of growth, development and stress response. The investigational design is well elaborated and makes use of appropriate, sophisticated techniques to achieve sound experimental evidence, which supports Authors' claims, through correct data interpretation and discussion.

The work appears of significant value in the specific field, its overall quality will certainly attract consideration in that context, and its message may extend to other experimental settings as well.

Reviewer #3 (Remarks to the Author):

Yuan et al identified that the pmk-1(D327E) mutation causes opposite phenotype to the loss-of-function of pmk-1 with ageing assay. Later on, the global gene expression profile further supported that the pmk-1(D327E) mutation likely leads to PMK-1 signaling up-regulation. The D327E mutation alters the putative CED-3 cleavage site. However, whether the CED-3-mediate cleavage is happening in vivo is not clear. The pmk-1(D327E) results into the reduction of total PMK-1 protein, while the phosphorylated PMK-1 seems remain at constant level as in wild type.

Major points:

1. This study did not provide sufficient evidence to support that the function of pmk-1(D327E) is related to CED-3-mediated cleavage. First of all, the existence of CED-3 cleavage product (PMK-1 CTA) has not been demonstrated in vivo. The authors argued that the cleavage product could be non-stable and degraded quickly in vivo. In Fig 5j, the PMK-1 CTA is visible after either proteasome or lysosome or autophagy inhibitor treatment. Similar treatment may be performed in ced-3 mutant to see whether the PMK-1 CTA level is decreased by ced-3. Alternatively, author may show that overexpressing CED-3 could enhance the PMK-1 CTA production.
2. The D327E mutation may prevent the production of PMK-1 CTA. However, another possibility is that the phosphorylated D327E-containing PMK-1 may have stronger activity in vivo. So far, the authors did not provide clear enough evidence to exclude this possibility. The authors may perform in vitro kinase assay to test whether PMK-1D327E has higher kinase activity than wild type PMK-1 or not.
3. CED-3 likely functions indirectly through PMK-1 phosphorylation but not direct cleavage to inhibit PMK-1 signaling. In consistent with this hypothesis, the phosphorylated PMK-1 is obviously increased in ced-3 mutant. Giving the significant up-regulation of active (phosphorylated) PMK-1 in ced-3, one can hardly believe that producing un-detectable level of non-functional PMK-1 CTA may do anything significantly to the pmk-1 signaling.
4. In pmk-1(D327E) mutant animals, the non-phosphorylated PMK-1 is reduced. Based upon, the authors thought that the relative amount of non-phosphorylated PMK-1 may play a key role in ageing

related function of PMK-1. Considering the complex signaling interactions and cross-talks among MAPKs, this assumption is somewhere farfetched. In general, without providing solid evidences to exclude the effect of phosphorylated PMK-1 on p38 signaling and feedback regulation from other MAPKs or other pathways, the current conclusion cannot hold.

5. The language in many places is vague and lacks clear explanations.

POINT-BY POINT RESPONSES

REVIEWER COMMENTS

Reviewer #1 (Remarks to the Author):

Yuan et al. characterize the consequences of altered PMK-1-p38 signaling in *C. elegans*. They use a mutant in the previously identified CED-3 cleavage site and a loss of function mutation. They report that loss of *pmk-1* reduces lifespan, contradicting a previous study that showed that in *daf-2* but not WT worms lifespan is significantly reduced by a *pmk-1* mutation. In contrast, the cleavage resistant mutant showed extended lifespan. Transcriptome experiments showed a differential impact of loss of function and cleavage resistant mutations in *pmk-1*. Based on these results they characterize lysosomes and subsequently neuronal integrity. Finally, they investigate the ratios of phosphorylated and total PMK-1 depending on the cleavage site.

The role of PMK-1 stress signaling in aging is generally interesting as it pertains to stress and immune signaling. The current manuscript suffers from lack of essential controls and the unstructured organization of the manuscript makes it very confusing and hard to follow.

We thank the reviewer for the comments to strengthen our claims and improve the manuscript.

Major concerns:

1. The degron strains are not sufficiently characterized. There are no controls for the degree of degradation of PMK-1 and, importantly, there is no control of the tissue specificity of the PMK-1 inactivation. This is an essential point because PMK-1 was previously shown to predominantly function in the intestine but endogenous PMK-1 protein has not been detected in any other cell type. This must be convincingly shown to support any of the non-cell autonomous claims of the paper. The degron strains must be carefully characterized before they are used as evidence. Moreover, no mechanism for non-cell autonomous roles are being explored. These findings are purely correlative.

We thank the reviewer for the comments on tissue specificity and address each in turn.

To address the degree of PMK-1 protein degradation by auxin induced degron (AID), we added a dose/time course characterization of auxin analog K-NAA. We find that at the K-NAA concentration of 0.1mM, which we used throughout the study, PMK-1 protein level was decreased to less than 5% within 18 hours of treatment (**New data in Supplementary Fig. 4a**).

To validate the tissue-specific degradation of PMK-1, we performed western blot to monitor the decrease of total PMK-1 and pPMK-1 protein levels under tissue-specific promoter driving TIR1 E3 Ligase with and without K-NAA auxin analog. We find that epidermis expresses a significant fraction of PMK-1 protein (**New data in Supplementary Fig. 4b**). Additionally, even given a situation where a tissue may have only a trace amount of PMK-1, the kinase may still have an important regulatory function.

Regarding PMK-1 functions in tissues other than intestine, work by the Ewbank and Pujol labs have previously shown that PMK-1 has important roles in epidermal pathogen responses ¹ making its epidermal roles in longevity further interesting.

Concerning the mechanism of cell non-autonomous function of PMK-1 regulating lysosome and PVD neuron integrity, our cell non-autonomous findings clarify the seemingly contradictory findings to a previous study that showed *pmk-1* mutation only affected lifespan in the presence of pathogen infection ². Importantly, previous studies were performed with the germline ablated. Our findings of *pmk-1* aging phenotypes are in the context of intact germline. Furthermore, we originally showed that germline *pmk-1* is essential for both epidermal lysosome formation and neuronal integrity (**Fig.3e and Fig. 4e**). We have **repeated the entire cell non-autonomy tests for both lysosome formation and neuronal integrity** and obtained the same results as our original findings (**New data pooled with previous data in Fig.3e and**

Fig.4e, individual experiments shown in Source Data). However, delineating the nature of the PMK-1 cross-tissue signal is beyond the scope of the questions addressed in this study. We have **added to the Discussion section** additional consideration of previous and current findings in the field with particular emphasis on neuronal proteostasis³⁻⁵ and non-cell autonomous signaling of p38 stress responses⁶⁻⁸.

2. It is important to show the consequences of the cleavage site mutant on pPMK-1/PMK-1 levels and the consequence on PMK-1 activity alongside the lifespan experiments in Figure 1. They are now shown at the end and for example Figure 6h is unclear. This panel is essential to substantiate the claims of the paper that the ratios of pPMK-1/PMK-1 play an important role in the activity of PMK-1. It appears that this heatmap is a conundrum of different data types and the original data need to be shown and carefully quantified.

We thank the reviewer for the comment on correlation of data presentation and for the question concerning how the heatmap in Fig. 6h was generated and address these issues in turn.

For the order of data presentation, we first present a striking phenotype that PMK-1, commonly known for stress responses, shortens the life span in the absence of stressors when germline is intact. Opposite to this, we find that PMK-1(D327E) mutation promotes life span extension—the mutant working opposite to the null is consistent with a gain-of-function. Based on these opposing effects on life span in the absence of stress, we chose to next examine the downstream pathways affected by eliminating PMK-1 function and whether these pathways are inversely impacted by PMK-1(D327E) mutation (**Fig 2**). In fact, the purpose Figure 2 is to reveal the consequence of cleavage resistant mutation on PMK-1 activity at a systems level. This “systems-to-targets” approach led us to the unexpected findings that PMK-1 regulated genes are enriched in lysosome and PVD neuronal components (**Fig 2**). We further investigated the impact of PMK-1 in these tissue and cellular compartments during aging (**Fig 3** and **Fig 4**). These findings at the cellular level provided additional evidence that the PMK-1 kinase has a role in maintaining tissue homeostasis during aging in the absence of stress. **Figure 5** addresses the question of what is the consequence of PMK-1(D327E) mutation at a molecular level revealing that phospho-ratio may be involved in signal strength. **Figure 6** further demonstrates that the non-phospho-PMK-1 is not a “silent partner” and its accumulation can attenuate PMK-1 signaling. Importantly, we provide several lines of evidence that PMK-1 signaling can be enhanced by lowering non-phospho-PMK-1 and that PMK-1 signaling can be lessened by increasing the non-phospho-PMK-1 pool. Finally, **Figure 6** further reveals that animals indeed have enhanced p38 signaling by decreasing non-phospho-PMK-1 during normal early aging process.

We thank the reviewer for bringing to our attention that the source of the data for the heatmap could have been clearer than what we stated in lines 447 – 450 of the original submission. With the intention of providing clarity, however, we have now **added a new section to the Methods (Estimating pPMK-1 levels during aging)** where we describe the rationale underlying our calculations and we also added a worksheet in the **Source Data Fig 6h** showing the step-by-step calculation. The heatmap (**Fig.6h**) is based on data from two measures. First, we estimated the relative changes in PMK-1 and relative changes in pPMK-1 based on the average of 3 independent western blots on Days 1, 5, and 10 of adulthood (shown in **Fig.5c-d** and **Supplementary Fig. 7**). Second, we determined the actual percent of pPMK-1 of total PMK-1 at Day 1 adulthood to be 5.7% as quantified by mass spectrometry (shown in **Fig. 5e**). We then set Day 1 wild type pPMK-1 to 5.7% and used the quantification of western blot to calculate the ratio of pPMK-1 to total PMK-1 for both wild type and PMK-1(D327E) on Day 1, 5 and 10.

3. Figures 2F, G are unclear. It appears there were two replicates that differ and this requires further repeats to solidify the data. The link to PMK-1 are unclear and the findings seems unconnected. The role of PMK-1 signaling in protein homeostasis remains unclear.

We thank the reviewer for the question about mass-spec measurements of K48 poly-ubiquitin conjugates (Previous Fig. 2f and g).

The two replicates in the previous Fig.2f-g are from two independent experiments with two separate mass spec runs. The antibody previously used for IP, Lys48-specific anti-Ubiquitin rabbit clone Apu2 (Sigma 05-1307), was discontinued by Sigma. To address the reviewer's concern, **we essentially repeated these mass-spec experiments with 3 more biological replicates which further validated our previous findings**. For these additional experiments, we used Polyubiquitin Enrichment Kit from Thermo Scientific (Cat# 89899) to enrich for polyubiquitinated proteins. We validated the pull down using both anti-ubiquitin antibody and Lys48-specific antibody (**New data in Supplementary Fig. 2c**). We performed additional 3 biological replicates for all 4 genotypes using the new Polyubiquitin Enrichment Kit (**New data in Fig. 2f-h**). To ensure repeatability, we considered a protein a "hit" if it appeared in **all three** replicates with $p < 0.05$ threshold.

Enrichment analysis of polyubiquitinated proteins inversely regulated by *pmk-1* null and *pmk-1(D327E)* (**New data in Fig. 2h**) showed results similar to what we originally obtained with Apu2 with significant GO term overlap with the original anti-Ubiquitin pull down experiments (**Previous Fig. 2g now Supplementary Fig.2f-h**), including extracellular components (cuticle), unfolded protein binding and ATP hydrolysis. Furthermore, the new results further confirmed that alteration of polyubiquitinated proteins in *pmk-1 (CTΔ)* is almost identical to *pmk-1(-)* (**New data in Fig.2f**), which is also consistent with our previous conclusion.

The goal of the experiments Figure 2 is to reveal, at a system level, what pathways are regulated by PMK-1 by investigating changes in both gene expression and protein degradation programs using both loss of function *pmk-1(-)* and gain of function *pmk-1(D327E)* mutation. We found that PMK-1 impacts a variety of downstream pathways of tissue homeostasis in addition to its previously established function in immune function. We chose to follow up on lysosome function and PVD neuron because they are most relevant to the aging phenotype shown in Figure 1.

4. How the lysosomes are regulated by the pPMK-1/PMK1 ratio remains mechanistically unclear. What is the role of the lysosomal aberrations in the consequence of *pmk-1* loss of function?

We thank the reviewer for the question concerning the role of pPMK-1/PMK-1 ratio in regulating lysosomes. In the previous version, we showed that PMK-1 signaling regulates lysosome function both in gene expression (Fig. 2a, 2d, Supplementary Fig.2b) and genetic analysis (Fig. 3, Fig. 6f). Now we have **new data** providing additional mechanism to support our conclusion (**New data in Supplementary Fig. 5b**).

Lysosomes are highly dynamic organelles constantly turning over, underscoring the importance of lysosome formation to maintain homeostasis^{9,10}. Previous studies have shown that enlargement of lysosomes is reflective of less functional lysosomes and long-lived mutants have youthful small lysosomes¹¹. To understand how *pmk-1(D327E)* extends lifespan whereas *pmk-1(-)* shortens lifespan, we identified about 100 genes inversely regulated. Lytic vacuole (lysosome) genes were one of highest enrichment categories (**Fig. 2b**). Based on the dysregulation of lytic genes by *pmk-1(-)* and the inverse effect with *pmk-1(D327E)* (**Fig. 2d**), we examined the lysosomal formation phenotype (**Fig. 3**). We find that lysosomes progressively enlarge in *pmk-1(-)* (**Fig. 3a-c**). Opposite to this, the cleavage-resistant *pmk-1(D327E)* extends a youthful phase of small lysosomes (**Fig. 3a-b**) suggesting a gain of function. We then did transcription factor analysis and revealed 8 more transcription factors in addition to *daf-16* that affect lysosome formation (**Fig. 3f**). Based on motif analysis, we proposed that some of the TFs may be regulated by PMK-1 signaling (**Fig. 3g**). In this revision, we **provided additional data** to show that *pmk-1* RNAi affects both total protein and phosphorylated forms of the top 3 transcription factors that had the most effect on lysosome formation (**New data in Supplementary Fig. 5b**).

As for phospho-ratio, we showed that *PMK-1(D327E)* mutation has enhanced pPMK-1/PMK-1 ratio (**Fig. 3a-d**) which leads to enhanced PMK-1 signaling (**Fig. 6j**). On the contrary, overexpressing wild type PMK-1 yields a decreased pPMK-1/PMK-1 ratio and shows the enlarged lysosome formation phenotype (**Fig. 6f**), consistent with diminished signaling. The findings of opposing functions by loss versus gain-of-function alleles is a powerful demonstration that the relative amount of PMK-1 signaling—as a result of the

differences in phospho-pool—regulates lysosome formation.

5. The link between PMK-1 and the transcription factors and lysosomal genes in Figure 3f are unclear. For example, is the activity of UNC-62, FOS-1 and JUN-1 in *pmk-1* null or degron mutants affected?

We thank the reviewer for the question how PMK-1 impacts these transcription factors. In the original version, we showed that these TFs were enriched for both proline-directed potential phosphorylation sites and MAPK docking motifs based on prediction of consensus sequences (**Fig. 3g** and **Supplementary Fig. 5a**). We now **provide an additional experiment** with FOS-1::GFP, JUN-1::GFP, and UNC-62::GFP fusion strains^{12, 13} to detect these TFs. We tested both mock and *pmk-1* RNAi and observed that *pmk-1* RNAi caused alterations in both phospho-forms and total TF protein accumulation for these TFs (**New data in Supplementary Fig. 5b**). In the original submission, we tested the impact of various TFs on lysosome formation using RNAi and found that these are the top 3 TFs that resulted in lysosome formation defects ablated (**Fig.3f**). Based on these new findings, it is reasonable that PMK-1 signaling regulates either phosphorylation or stability of these transcription factors. Thus, our main conclusion for this section remains that a complex transcriptional network of multiple TFs impacts lysosome formation.

6. The connection between lifespan, lysosomes and pain sensory neurons is unclear.

We thank the reviewer for the comment on tissue structures and lifespan. Importantly, loss of tissue integrity is a hallmark of aging and major source of debility. Yet, understanding the underlying defects remains a research challenge with few examples shown to improve tissue integrity. Beyond degradation of macromolecules, lysosomes are emerging as key to multiple aspects of homeostasis^{9, 10, 14, 15}. Moreover, emerging findings suggest complex roles for lysosome activities supporting lifespan extension^{11, 16-19}. Finally, compromised lysosome function is thought to be a major factor contributing to all forms of neurodegeneration²⁰ and promoting lysosome formation has only very recently been speculated as feasible for prevention of neurodegeneration¹⁵.

Our findings of the PMK-1 function in lysosome formation during development and aging allows us an entry point to deeply understand the interplay of physiological and molecular mechanisms in tissue integrity. It is not uncommon for mutants to have an array of defects. However, the opposing outcomes of *pmk-1* null with *pmk-1(D327E)* provides a powerful demonstration of specificity that the relative signaling strength of PMK-1 controls tissue homeostasis. Our findings that the relative extent of PMK-1 signaling supports tissue homeostasis including lysosome formation and integrity of pain sensory neurons is very timely. We have **added a new Discussion section** to consider the context of our findings with previous findings.

7. The role of the cleavage site *in vivo* remains rather unclear based on the results presented in Figure 5. For instance epistasis experiments in Figure 1 with *ced-3* and other outcomes of PMK-1 signaling would help clarifying this.

We thank the reviewer for the comment on the role of PMK-1 cleavage *in vivo*. We have now **added new data** to show that in *ced-3(-)* null mutants, numerous PMK-1 target genes are up-regulated suggesting *ced-3(-)* animals have enhanced p38 signaling. (**New data in Fig. 5n**).

In Figure 5, we now show **3 different** demonstrations that **CED-3 caspase cleavage limits pPMK-1 levels**. First, compared to wild-type, the CED-3 cleavage resistant PMK-1(D327E) mutation led to increased pPMK-1 when organismal adaptation is eliminated or circumvented (**Fig. 5f-g** for overexpression and **Fig. 5h-i** for transient de-repression). Second, bortezomib treatment to block the proteasome led to increased pPMK-1 accumulation for both *pmk-1(D327E)* and *ced-3(-)* mutants compared to wild-type (**Fig. 5l-m**). Third, both *pmk-1(D327E)* and *ced-3(-)* mutants have increased PMK-1 downstream target gene expression (**New data in Fig. 5n** and **Fig. 6j**). These results are also consistent with the enhanced signaling for PMK-1(D327E) in life span (**Fig. 1**), lysosome formation (**Fig. 3**) and neuronal integrity (**Fig. 4**) phenotypes.

8. Numbers of experimental repeats etc. are lacking throughout.

We thank the reviewer for pointing out this issue. We have more clearly marked replicate values in the text and legends. We have presented replicates of key experiments in supplementary figures and provided source data for all experiments.

Additional concerns:

1. 128-130: The cleaved protein is not functional in the context of lifespan. Can it have other functions?

That is certainly possible; however, for each phenotype we have measured, we find that *pmk-1* CTD functions like the null including life span (**Fig. 1e**), polyubiquitinated protein enrichments (**Fig. 2f**), lysosome formation (**Supplementary Fig 3a-b**), neuronal integrity (**Supplementary Fig. 6c-d**), protein expression (**Supplementary Fig. 8b**) and growth rate (**Supplementary Fig 9a**).

2. 143-144: They claim that cleavage resistant *pmk-1* could not restore *daf-16* mutants to normal lifespan. There is no wt control neither in figure 1f nor in Supplementary 1d. Was this lifespan performed in parallel to the one in 1e?

We apologize for the confusion. We have now provided a replicate of this aging assay with wild-type animal measured at the same time (**New data in Supplementary Fig. 1e**) showing *pmk-1(D327E)* did not restore *daf-16(-)* to normal life span. This is also a consistent observation based on average life span of wild type animals across different experiments.

3. 162-164 “ Most genes altered by loss of *pmk-1* are also affected by loss of *daf-16*, suggesting PMK-1 signaling and DAF-16 co-regulate these genes”: reference? Is this known also for genes other than the ones involved in innate immune response?

We apologize for the confusion. This conclusion was referring to our observation for PVD and Lysosome genes (**Fig .2c-d**). We show that DAF-16 does impact expression of PMK-1 dependent PVD and lysosomal genes. However, loss of *daf-16* does not alter the inverse regulation of these genes by PMK-1 vs PMK-1 (D327E), suggesting that there are additional TFs working downstream of PMK-1 to control the expression of these genes.

4. Supplementary Fig 2: In a and b are the heatmaps compared to wt? Expand legends. All of our expression values are relative to wild-type. This has been stated more clearly in the figure legends.

5. Fig. 2f: controls are missing.

We apologize for the confusion. We have replaced Fig.2f with a Venn diagram showing alteration in all genotypes relative to wild type to clarify.

6. Fig. 3a: how are the tubules and particulates defined?

For quantification of lysosomal structure, puncta with circularity of 0.4 - 1 was quantified as particles and puncta with circularity of 0 - 0.4 was quantified as tubules. This has been added to the methods.

7. Fig. 3c: If the lysosome enlargement is happening from L1 to L4, is it a developmental phenotype or an ageing phenotype?

This phenotype has both developmental and aging components and it is progressive with age.

8. Fig 3e: Why do neurons and intestine have no effect?

Our data suggest that epidermal lysosomes are most impacted by epidermal and germline PMK-1. This should not be taken to mean that intestinal PMK-1 has no effect—just not as easily detected for epidermal lysosomes.

9. Fig. 3f: Can it be that the transcription factors KD is affecting particle size independently of *pmk-1*? 258-259: The suggestions that PMK-1 signals through a complex transcriptional network to regulate lysosomal genes is not well supported.

We thank the reviewer for this comment. Our data show that *pmk-1*-dependent genes are highly enriched for lysosomal genes. Analyzing the promoters of these genes have revealed a dozen TFs binding to their promoters (**Figure 2e**) and that some of these TFs do impact lysosome formation (**Figure 3f**). We now **provide an additional experiment** showing *pmk-1* RNAi caused alterations in both phospho-forms and total TF protein accumulation for the top three TFs that impact lysosome formation (**New data in Supplementary Fig. 5b**). Based on all these findings, PMK-1 signaling likely regulates either phosphorylation or stability of these transcription factors relevant to lysosome formation.

Reviewer #2 (Remarks to the Author):

The paper provides interesting knowledge about the complex function of PMK-1 enzyme family in *C. elegans* tissue homeostasis and longevity, in particular describing the critical role of non-phospho-p38 stoichiometry in the proper balance of growth, development and stress response. The investigational design is well elaborated and makes use of appropriate, sophisticated techniques to achieve sound experimental evidence, which supports Authors' claims, through correct data interpretation and discussion. The work appears of significant value in the specific field, its overall quality will certainly attract consideration in that context, and its message may extend to other experimental settings as well.

We thank the reviewer for the overall strong and positive comments.

Reviewer #3 (Remarks to the Author):

Yuan et al identified that the *pmk-1*(D327E) mutation causes opposite phenotype to the loss-of-function of *pmk-1* with ageing assay. Later on, the global gene expression profile further supported that the *pmk-1*(D327E) mutation likely leads to PMK-1 signaling up-regulation. The D327E mutation alters the putative CED-3 cleavage site. However, whether the CED-3-mediate cleavage is happening in vivo is not clear. The *pmk-1*(D327E) results into the reduction of total PMK-1 protein, while the phosphorylated PMK-1 seems remain at constant level as in wild type.

We thank the reviewer for the comments to strengthen our claims and improve the manuscript.

Major points:

1. This study did not provide sufficient evidence to support that the function of *pmk-1*(D327E) is related to CED-3-mediated cleavage. First of all, the existence of CED-3 cleavage product (PMK-1 C Δ) has not been demonstrated in vivo. The authors argued that the cleavage product could be non-stable and degraded quickly in vivo. In Fig 5j, the PMK-1 C Δ is visible after either proteasome or lysosome or autophagy inhibitor treatment. Similar treatment may be performed in *ced-3* mutant to see whether the PMK-1 C Δ level is decreased by *ced-3*. Alternatively, author may show that overexpressing CED-3 could enhance the PMK-1 C Δ production.

We thank the reviewer for the comments on the specificity of caspase cleavage of PMK-1. We previously established that CED-3 caspase cleaves PMK-1 specifically at Asp327 and not at any of several other Asp

residues²¹. To clarify the Fig. 5j experiment, PMK-1 (CTΔ) band is from PMK-1 (CTΔ) mutant generated using CRISPR mutagenesis. This was done as we cannot observe any direct cleavage product under normal western blot conditions.

We reasoned that the cleaved PMK-1 product is unstable and targeted to degradation quickly. Consistent with this, the amount of PMK-1 (CTΔ) mutant protein from PMK-1 (CTΔ) mutant animals is very low compared to wild type PMK-1 *in vivo* (**Supplementary Fig. 8b**) despite mRNA levels similar to wild-type *pmk-1* (**Supplementary Fig. 8c**). However, the PMK-1 (CTΔ) protein has similar protein stability *in vitro* as wild type PMK-1 (**Supplementary Fig. 8d**), supporting PMK-1 (CTΔ) is quickly turned over *in vivo*. Therefore, we are unable to evaluate how *ced-3(-)* affects the accumulation of cleavage product because we cannot observe it directly. Unfortunately, overexpressing CED-3 is sufficient to kill any cell type, making the suggestion technically unfeasible.

Nonetheless, to address the reviewer's concern about whether the function of PMK-1 is affected by CED-3 directly *in vivo*, we **added new data** to show that in *ced-3(-)* animals, PMK-1 target genes are up-regulated suggesting *ced-3(-)* animals have enhanced p38 signaling (**New data in Fig. 5n**).

2. The D327E mutation may prevent the production of PMK-1 CTΔ. However, another possibility is that the phosphorylated D327E-containing PMK-1 may have stronger activity *in vivo*. So far, the authors did not provide clear enough evidence to exclude this possibility. The authors may perform *in vitro* kinase assay to test whether PMK-1D327E has higher kinase activity than wild type PMK-1 or not.

We appreciate the reviewer's concern on the impact of an Asp or Glu acidic residue at the 327 position in PMK-1 and their relative impacts on signaling strength. However, it is well-established that either acidic residue in this position does not impact kinase activity. As examples, human p38 alpha and p38 delta contain Asp and Glu residues, respectively, in the same position as Asp327 in PMK-1 (**Fig 1a**) and have comparable kinase activities. Moreover, many other MAPKs such as ERK also have a Glu in the same position.

3. CED-3 likely functions indirectly through PMK-1 phosphorylation but not direct cleavage to inhibit PMK-1 signaling. In consistent with this hypothesis, the phosphorylated PMK-1 is obviously increased in *ced-3* mutant. Giving the significant up-regulation of active (phosphorylated) PMK-1 in *ced-3*, one can hardly believe that producing un-detectable level of non-functional PMK-1 CTΔ may do anything significantly to the *pmk-1* signaling.

We thank the reviewer for the comment on CED-3 cleavage of PMK-1 and connection to PMK-1 CTΔ and address each in turn.

As the reviewer noted, phosphorylated PMK-1 is markedly increased in *ced-3* mutants with proteasome inhibition by about 3-fold compared to wild-type. Importantly the phosphorylated PMK-1 is also increased in *pmk-1(D327E)*---the cleavage resistant mutant---by more than 3-fold with proteasome inhibition compared to wild-type (**Fig 5l-m** and **Supplementary Fig 8e**). This is a strong demonstration of specificity from both enzyme (CED-3) and substrate (PMK-1) because in the *ced-3(-)* mutant, PMK-1 is intact and in PMK-1(D327E) mutant, CED-3 is functional. Yet in **both** mutants we find the same result---increased pPMK-1 accumulation.

As for the contribution of "undetectable levels" of PMK-1 CTΔ, we would like to clarify. The 3-fold upregulation of pPMK-1 in both *ced-3(-)* and *pmk-1(D327E)* did not impact the level of total PMK-1 protein (**Fig 5l-m** and **Supplementary Fig 8e**) because pPMK-1 is a very small fraction of total PMK-1 as we showed with mass spectrometry (**Fig 5e**) with only 1 out of 20 molecules phosphorylated in the absence of stress response. Yet the pPMK-1 is essential for signaling despite the low level. Therefore, it is very likely that slower turnover of the pPMK-1 in *ced-3(-)* mutant enhances signaling.

Once cleaved, the truncated fragments are either stable or rapidly destabilized as degrons as we previously defined for LIN-28²². Additionally, to date, identifying non-apoptotic targets remains one of the greatest challenges in the field because---for exactly as the reviewer has pointed out here that---cleavage products have been quite elusive *in vivo* given their rapid destabilization following cleavage. Although counterintuitive at first, it makes a lot of sense because in the context of live cells, caspase functions are non-lethal and if the caspase activity was elevated sufficiently to make the cleavage products more obvious *in vivo*, this would result in cell death (ie. high expression of caspase is sufficient to kill cells).

4. In pmk-1(D327E) mutant animals, the non-phosphorylated PMK-1 is reduced. Based upon, the authors thought that the relative amount of non-phosphorylated PMK-1 may play a key role in ageing related function of PMK-1. Considering the complex signaling interactions and cross-talks among MAPKs, this assumption is somewhere farfetched. In general, without providing solid evidences to exclude the effect of phosphorylated PMK-1 on p38 signaling and feedback regulation from other MAPKs or other pathways, the current conclusion cannot hold.

We thank the reviewer for the comments on the effect of non-phosphorylated PMK-1. In fact, we did show that overexpression of phospho-site dead PMK-1 (TGY to AGF) has the same effect as overexpression of wild-type PMK-1---both rescued the larval stall caused by *vhp-1(RNAi)* induced PMK-1 hyper-phosphorylation (**Supplementary Fig. 9c-d**). These findings directly support the conclusion that non-phosphorylated PMK-1 is able to modulate PMK-1 signaling.

To clarify, we are not arguing against the importance of pPMK-1 in p38 signaling. On the contrary, the amount of pPMK-1 in animals is very important because it is strictly regulated (**Fig.5a-d** and **Fig. 6d**). Moreover, hyper-activation of pPMK-1 results in developmental stall (**Supplementary Fig. 9a**). Therefore, given the important function of p38 kinase in supporting tissue homeostasis, it is crucial to regulate PMK-1 signaling at homeostatic level without hyperactivation. As such, our findings strongly support the conclusion that non-phosphorylated PMK-1 is not a “silent partner” and its relative fraction tunes overall signaling up or down as it is decreased or increased, respectively. These findings provide an additional mechanism to modulate p38 signaling without changing pPMK-1 levels.

Regulation of MAPK signaling in stress responses is well-established to activate diverse mediators and involve feedback²³⁻²⁵. Whether or not non-phosphorylated PMK-1 impacts cross-talk with other MAPKs does not alter the findings for the question we were addressing. That part of our study addresses the impact of non-phospho-PMK-1 to PMK-1 signaling. Addressing other MAPK crosstalk is an intriguing but open-ended question and others may also be stimulated to test such interactions for MAPKs of their own particular interest.

5. The language in many places is vague and lacks clear explanations.

We thank the reviewer for this comment and have worked hard to edit the manuscript and legends to improve clarity.

References

1. Zugasti, O. *et al.* A quantitative genome-wide RNAi screen in *C. elegans* for antifungal innate immunity genes. *BMC. Biol* **14**, 35 (2016).
2. Troemel, E.R. *et al.* p38 MAPK regulates expression of immune response genes and contributes to longevity in *C. elegans*. *PLoS Genet* **2**, e183 (2006).
3. Ding, C., Wu, Y., Dabas, H. & Hammarlund, M. Activation of the CaMKII-Sarm1-ASK1-p38 MAP kinase pathway protects against axon degeneration caused by loss of mitochondria. *Elife* **11** (2022).
4. Park, E.C. & Rongo, C. The p38 MAP kinase pathway modulates the hypoxia response and glutamate receptor trafficking in aging neurons. *Elife* **5** (2016).

5. Toth, M.L. *et al.* Neurite sprouting and synapse deterioration in the aging *Caenorhabditis elegans* nervous system. *J Neurosci* **32**, 8778-8790 (2012).
6. Sun, J., Singh, V., Kajino-Sakamoto, R. & Aballay, A. Neuronal GPCR controls innate immunity by regulating noncanonical unfolded protein response genes. *Science* **332**, 729-732 (2011).
7. Foster, K.J. *et al.* Innate Immunity in the *C. elegans* Intestine Is Programmed by a Neuronal Regulator of AWC Olfactory Neuron Development. *Cell Rep* **31**, 107478 (2020).
8. Otarigho, B. & Aballay, A. Immunity-longevity tradeoff neurally controlled by GABAergic transcription factor PITX1/UNC-30. *Cell Rep* **35**, 109187 (2021).
9. Luzio, J.P., Pryor, P.R. & Bright, N.A. Lysosomes: fusion and function. *Nat Rev Mol Cell Biol* **8**, 622-632 (2007).
10. Ballabio, A. & Bonifacino, J.S. Lysosomes as dynamic regulators of cell and organismal homeostasis. *Nat Rev Mol Cell Biol* **21**, 101-118 (2020).
11. Sun, Y. *et al.* Lysosome activity is modulated by multiple longevity pathways and is important for lifespan extension in *C. elegans*. *Elife* **9** (2020).
12. Sarov, M. *et al.* A recombineering pipeline for functional genomics applied to *Caenorhabditis elegans*. *Nat Methods* **3**, 839-844 (2006).
13. Gerstein, M.B. *et al.* Integrative analysis of the *Caenorhabditis elegans* genome by the modENCODE project. *Science* **330**, 1775-1787 (2010).
14. Chen, Y. & Yu, L. Scissors for autolysosome tubules. *EMBO J* **34**, 2217-2218 (2015).
15. Zoncu, R. & Perera, R.M. Built to last: lysosome remodeling and repair in health and disease. *Trends Cell Biol* **32**, 597-610 (2022).
16. Lapiere, L.R. *et al.* The TFEB orthologue HLH-30 regulates autophagy and modulates longevity in *Caenorhabditis elegans*. *Nat Commun* **4**, 2267 (2013).
17. Folick, A. *et al.* Aging. Lysosomal signaling molecules regulate longevity in *Caenorhabditis elegans*. *Science* **347**, 83-86 (2015).
18. Silvestrini, M.J. *et al.* Nuclear Export Inhibition Enhances HLH-30/TFEB Activity, Autophagy, and Lifespan. *Cell Rep* **23**, 1915-1921 (2018).
19. Savini, M. *et al.* Lysosome lipid signalling from the periphery to neurons regulates longevity. *Nat Cell Biol* **24**, 906-916 (2022).
20. Udayar, V., Chen, Y., Sidransky, E. & Jagasia, R. Lysosomal dysfunction in neurodegeneration: emerging concepts and methods. *Trends Neurosci* **45**, 184-199 (2022).
21. Weaver, B.P. *et al.* Non-Canonical Caspase Activity Antagonizes p38 MAPK Stress-Priming Function to Support Development. *Dev Cell* **53**, 358-369 e356 (2020).
22. Weaver, B.P., Weaver, Y.M., Mitani, S. & Han, M. Coupled Caspase and N-End Rule Ligase Activities Allow Recognition and Degradation of Pluripotency Factor LIN-28 during Non-Apoptotic Development. *Dev. Cell* **41**, 665-673 (2017).
23. Canovas, B. & Nebreda, A.R. Diversity and versatility of p38 kinase signalling in health and disease. *Nat Rev Mol Cell Biol* **22**, 346-366 (2021).
24. Balasubramaniam, B. *et al.* p38-MAPK recruits the proteolytic pathways in *Caenorhabditis elegans* during bacterial infection. *Int J Biol Macromol* **204**, 116-135 (2022).
25. Rajpoot, S., Kumar, A., Zhang, K.Y.J., Gan, S.H. & Baig, M.S. TIRAP-mediated activation of p38 MAPK in inflammatory signaling. *Sci Rep* **12**, 5601 (2022).

REVIEWERS' COMMENTS

Reviewer #1 (Remarks to the Author):

In the revised manuscript, Yuan et al. explain better the connection between their experiments and their claims. The flow of the manuscript has been substantially improved and the authors have included many of the controls that were asked and have strengthened their hypothesis. They have also expanded their figure legends to make them more understandable.

There are, however, unresolved issues:

Major

1. The authors have attempted to resolve the essential reviewer point of showing tissue specificity of their knockout strains by using western blotting of whole worm extracts. However, this method is simply unsuitable to show the cell type specificity. This could only be shown by IF staining or, if the authors insist on WB, possibly by (more complicated) WB of sorted cells but not whole worm extracts. The "tissue-specific promoter driving TIR1 E3 Ligase" is only an indirect argument, because there could be leakiness of the constructs. Therefore, the interpretation of the tissue specific function of PMK-1 is not convincingly demonstrated.

Minor

1. Supplementary Fig. 5b shows a convincing experiment to validate the changes in the phosphorylation levels of PMK-1 potential substrates. Have these experiments been repeated and were findings reproduced?

2. Missing from materials and methods: Supplementary Fig. 5b procedure should be further explained (use of Phos-Tag gel), as well as how the tagged FOS-1, JUN-1 and UNC-62 tagged proteins were generated or acquired.

Reviewer #3 (Remarks to the Author):

Issues raised earlier are reasonably addressed. The general conclusions are sensible.

REVIEWERS' COMMENTS

Reviewer #1 (Remarks to the Author):

In the revised manuscript, Yuan et al. explain better the connection between their experiments and their claims. The flow of the manuscript has been substantially improved and the authors have included many of the controls that were asked and have strengthened their hypothesis. They have also expanded their figure legends to make them more understandable.

There are, however, unresolved issues:

Major

1. The authors have attempted to resolve the essential reviewer point of showing tissue specificity of their knockout strains by using western blotting of whole worm extracts. However, this method is simply unsuitable to show the cell type specificity. This could only be shown by IF staining or, if the authors insist on WB, possibly by (more complicated) WB of sorted cells but not whole worm extracts. The “tissue-specific promoter driving TIR1 E3 Ligase” is only an indirect argument, because there could be leakiness of the constructs. Therefore, the interpretation of the tissue specific function of PMK-1 is not convincingly demonstrated.

The reviewer's concern is noted. However, we submit that the point has been satisfactorily addressed both within the literature and use of state-of-the-art approaches within this study. We provide the following rationale.

First, regarding TIR1 expression in specific tissues, the promoters we used have been explicitly recommended for use with TIR1 E3 ligase as they have already been validated by a collaboration of half a dozen other *C. elegans* labs with expertise on this topic ^{1,2}.

Second, the set of tissue-specific promoters we have used were chosen based on their well-documented use as high-fidelity promoters in *C. elegans*. Previous studies that validated the specificity of these promoters at the same stages analyzed in our study include: the ***eft-3***(*eef-1A.1*) promoter for whole body expression ³⁻⁵, the ***col-10*** promoter for epidermal (hypodermal) expression ^{6,7}, the ***ges-1*** promoter for intestine-specific expression ^{8,9}, the ***rgef-1*** promoter for pan-neuronal expression ^{10,11}, the ***myo-3*** promoter for body wall muscle-specific expression ¹²⁻¹⁴, and the ***sun-1***(*mtf-1*) promoter for germline-specific expression ¹⁵⁻¹⁹.

Third, these promoters have also shown fidelity with multi-copy arrays. Important to this point, we only used single copy insertions in Mos1 loci validated for fidelity of expression ^{3,20}. Single copy insertions in these validated Mos1 sites further limits excessive expression and prevents mosaicism (stochastic silencing).

Fourth, immunofluorescence does not work for staining PMK-1 in this case as we have tried multiple times with no success. We previously made a PMK-1::GFP fusion by CRISPR mutagenesis. We observe the fusion product at the right size by Western blot but could not see the fusion GFP *in vivo* either by direct fluorescence or by immuno-staining. Altogether, observing PMK-1 *in vivo* remains technically unfeasible with current technologies. Moreover, TIR-1 expression by fusion to mRuby with and without auxin degradation has already been shown ².

Fifth, sorting cells to separate tissues is not something standard in *C. elegans* and is quite specialized due to difficulties with the strong collagen cuticle. Moreover, this approach would have exactly the same caveat because sorting would have to be based on tissue-specific expression of a fluorescent marker *using the same promoters*.

Finally, we understand the reviewer's concern that any tissue-specific promoter could be “leaky” in principle. By definition, any molecular probing approach or tissue-specific functional assay will come with caveats. Nonetheless, we have now strengthened our rationale within the main text (modified both **Results** and **Methods** sections) as well as **added key references** to the study. We have also added a discussion point on tissue specificity (modified **Discussion**).

Minor

1. Supplementary Fig. 5b shows a convincing experiment to validate the changes in the phosphorylation levels of PMK-1 potential substrates. Have these experiments been repeated and were findings reproduced?
2. Missing from materials and methods: Supplementary Fig. 5b procedure should be further explained (use of Phos-Tag gel), as well as how the tagged FOS-1, JUN-1 and UNC-62 tagged proteins were generated or acquired.

We thank the reviewer for both of these minor points. We have clarified the experimental details (revised **methods**) and provided a statement of single experiment in the legend (**Supplementary Fig. 5**). The strain details were previously listed in the table of all strains used in this study (**Supplementary Data 5**) and the **references** for these strains have been added (revised **methods**).

Reviewer #3 (Remarks to the Author):

Issues raised earlier are reasonably addressed. The general conclusions are sensible.

We thank the reviewer.

References

1. Ashley, G.E. *et al.* An expanded auxin-inducible degron toolkit for *Caenorhabditis elegans*. *Genetics* **217** (2021).
2. Zhang, L., Ward, J.D., Cheng, Z. & Dernburg, A.F. The auxin-inducible degradation (AID) system enables versatile conditional protein depletion in *C. elegans*. *Development* **142**, 4374-4384 (2015).
3. Frokjaer-Jensen, C., Davis, M.W., Ailion, M. & Jorgensen, E.M. Improved Mos1-mediated transgenesis in *C. elegans*. *Nat Methods* **9**, 117-118 (2012).
4. Frokjaer-Jensen, C. *et al.* An Abundant Class of Non-coding DNA Can Prevent Stochastic Gene Silencing in the *C. elegans* Germline. *Cell* **166**, 343-357 (2016).
5. Goudeau, J. *et al.* Split-wrmScarlet and split-sfGFP: tools for faster, easier fluorescent labeling of endogenous proteins in *Caenorhabditis elegans*. *Genetics* **217** (2021).
6. Hong, Y., Lee, R.C. & Ambros, V. Structure and function analysis of LIN-14, a temporal regulator of postembryonic developmental events in *Caenorhabditis elegans*. *Mol Cell Biol* **20**, 2285-2295 (2000).
7. Yoder, J.H., Chong, H., Guan, K.L. & Han, M. Modulation of KSR activity in *Caenorhabditis elegans* by Zn ions, PAR-1 kinase and PP2A phosphatase. *EMBO J* **23**, 111-119 (2004).
8. Aamodt, E.J., Chung, M.A. & McGhee, J.D. Spatial control of gut-specific gene expression during *Caenorhabditis elegans* development. *Science* **252**, 579-582 (1991).
9. Edgar, L.G. & McGhee, J.D. Embryonic expression of a gut-specific esterase in *Caenorhabditis elegans*. *Dev Biol* **114**, 109-118 (1986).
10. Altun-Gultekin, Z. *et al.* A regulatory cascade of three homeobox genes, *ceh-10*, *ttx-3* and *ceh-23*, controls cell fate specification of a defined interneuron class in *C. elegans*. *Development* **128**, 1951-1969 (2001).
11. Chen, L., Fu, Y., Ren, M., Xiao, B. & Rubin, C.S. A RasGRP, *C. elegans* RGEF-1b, couples external stimuli to behavior by activating LET-60 (Ras) in sensory neurons. *Neuron* **70**, 51-65 (2011).
12. Fire, A. & Waterston, R.H. Proper expression of myosin genes in transgenic nematodes. *EMBO J* **8**, 3419-3428 (1989).
13. Ahier, A. *et al.* Affinity purification of cell-specific mitochondria from whole animals resolves patterns of genetic mosaicism. *Nat Cell Biol* **20**, 352-360 (2018).
14. Vrablik, T.L., Wang, W., Upadhyay, A. & Hanna-Rose, W. Muscle type-specific responses to NAD⁺ salvage biosynthesis promote muscle function in *Caenorhabditis elegans*. *Dev Biol* **349**, 387-394 (2011).
15. Malone, C.J. *et al.* The *C. elegans* hook protein, ZYG-12, mediates the essential attachment between the centrosome and nucleus. *Cell* **115**, 825-836 (2003).

16. Fridkin, A. *et al.* Matefin, a *Caenorhabditis elegans* germ line-specific SUN-domain nuclear membrane protein, is essential for early embryonic and germ cell development. *Proc Natl Acad Sci U S A* **101**, 6987-6992 (2004).
17. Penkner, A. *et al.* The nuclear envelope protein Matefin/SUN-1 is required for homologous pairing in *C. elegans* meiosis. *Dev Cell* **12**, 873-885 (2007).
18. Daryabeigi, A. *et al.* Nuclear Envelope Retention of LINC Complexes Is Promoted by SUN-1 Oligomerization in the *Caenorhabditis elegans* Germ Line. *Genetics* **203**, 733-748 (2016).
19. Kim, H.J., Liu, C., Zhang, L. & Dernburg, A.F. MJL-1 is a nuclear envelope protein required for homologous chromosome pairing and regulation of synapsis during meiosis in *C. elegans*. *Sci Adv* **9**, eadd1453 (2023).
20. Frokjaer-Jensen, C. *et al.* Single-copy insertion of transgenes in *Caenorhabditis elegans*. *Nat. Genet* **40**, 1375-1383 (2008).